# Protective Factors for LGBTI+ Youth Wellbeing: A Scoping Review Underpinned by Recognition Theory

**DOI:** 10.3390/ijerph182111682

**Published:** 2021-11-07

**Authors:** Nerilee Ceatha, Aaron C. C. Koay, Conor Buggy, Oscar James, Louise Tully, Marta Bustillo, Des Crowley

**Affiliations:** 1School of Social Policy, Social Work and Social Justice, University College Dublin, Belfield, Dublin 4, Ireland; 2Department of Sociology, Trinity College Dublin, Dublin 2, Ireland; cheechek@tcd.ie; 3School of Sociology, University College Dublin, Belfield, Dublin 4, Ireland; 4School of Public Health, Physiotherapy and Sports Science, University College Dublin, Belfield, Dublin 4, Ireland; conor.buggy@ucd.ie; 5Independent Researcher, Dublin, Ireland; oscar.ben.james@gmail.com; 6School of Physiotherapy, Division of Population Health Sciences, RCSI University of Medicine and Health Sciences, Dublin 2, Ireland; LouiseTully@rcsi.ie; 7UCD Library, University College Dublin, Belfield, Dublin 4, Ireland; marta.bustillo@ucd.ie; 8School of Medicine, University College Dublin, Belfield, Dublin 4, Ireland; doctordes@hotmail.com; 9HSE Addiction Services, Health Services Executive, Dublin 11, Ireland; 10Addiction Management in Primary Care, Irish College of General Practitioners, Dublin 2, Ireland

**Keywords:** youth, LGBTI+, wellbeing, Recognition Theory, scoping review, protective factors, sexual minority youth (SMY), gender minority youth (GMY), diverse sex development, intersex

## Abstract

Considerable research has been undertaken regarding the mental health inequalities experienced by lesbian, gay, bisexual, transgender and intersex (LGBTI+) youth as a consequence of societal and individual prejudice, stigma and discrimination. Far less research has focussed on protective factors that promote wellbeing for this population. A scoping review was conducted using a six-stage methodological framework, and is reported in accordance with the PRISMA-ScR statement. This explored the extent, range and nature of the peer-reviewed, published, academic literature on what is known about the protective factors that promote LGBTI+ youth wellbeing. Six databases were systematically searched applying Population–Concept–Context key inclusion criteria, complemented by contact with authors to identify additional sources, reference checks and hand searches. Ninety-six individual research records were identified and analysed, drawing from Honneth’s Recognition Theory. Interpersonal relations with parents (*n* = 40), peers (*n* = 32) and providers (*n* = 22) were associated with indicators of enhanced wellbeing, as were LGBTI+ community relations (*n* = 32). Importantly, online (*n* = 10), faith (*n* = 10) and cultural (*n* = 5) communities were potentially protective. Content and thematic analysis highlighted the importance of Gay–Straight Alliances (GSAs) (*n* = 23) offering powerful protective opportunities through intersecting interpersonal, community and legal forms of recognition. GSAs enhance allyship by peers and providers (*n* = 21), facilitate access to LGBTI+ community networks (*n* = 11) and co-exist alongside inclusive policies (*n* = 12), curricular (*n* = 5) and extracurricular activities (*n* = 1). This scoping review underscores the need to move beyond the predominant focus on risk factors for LGBTI+ youth, which subsequently inform protectionist approaches. It concludes with an appeal to develop mechanisms to apply recognitive justice to policy, practice and, importantly, future research directions. This emphasises the salience of enhanced understandings of inclusion, which is rights-based, universally available and of potential benefit to all.

## 1. Introduction

### 1.1. Orientations and Identities 

The World Health Organization (WHO) describes mental health holistically as “a state of wellbeing” [1]. While the concept of wellbeing is contested, it is used extensively throughout the literature, with less clarity about how this is defined [2,3]. In relation to lesbian, gay, bisexual, transgender and intersex (LGBTI+) populations, the concept of wellbeing has generated considerable research interest over many decades [4]. Defining sexual minority, gender minority youth and those with diverse sex development using the LGBTI+ acronym appears straightforward: “L” equates to lesbian; “G” to gay; “B” to bisexual; “T” to transgender and “I” to intersex. The LGBTI+ acronym comprises three dimensions, sexual orientation, gender identity and sex development, with wide variations and diversity of expression, particularly for youth [5]. Sexual orientation encompasses identification, behaviour and attraction, with suggestions of a greater lifetime prevalence of same-gender behaviour and attraction than identification [6], and higher prevalence of an LGBTI+ identification for youth, with young people more likely to identify as bisexual [7]. Gender identity refers to someone’s internal sense of their gender as male, female or non-binary, and may not accord with the sex assigned at birth [4]. Sex development is a spectrum of variations that occur within humanity, including intersex youth [8]. The inclusion of populations with diverse sex development accords with the recent work of the National Academies of Sciences, Engineering and Medicine [9]. Notwithstanding the complexity of youth orientations and identities, “researchers tend to use self-identification as the defining criterion” [4] (p. 13).

### 1.2. Wellbeing and Stigmatisation

There has been much concern regarding youth mental health disparities and vulnerability to psychological distress and suicidality [4,10,11,12]. This is typically contextualised within a Minority Stress Model, which describes the consequences of discrimination against, and victimisation of, marginalised groups [13]. This is consistent with WHO identification of the negative impact of social exclusion and stigmatisation [14]. Further, structural stigma within systems and enactment of personal stigma at the intersubjective level are acknowledged [15]. Stigmatisation regarding LGBTI+ identities is recognised as impacting negatively on wellbeing, reinscribing normative, binary frames of reference [16]. Such stigmatisation may equally apply to “mental health”, regarded as synonymous with ill health, as distinct from positive mental health or social wellbeing [2]. The resultant tendency for young LGBTI+ lives to be represented as universally vulnerable and “at risk”, on the basis of their orientations and identities, may lead to protectionist approaches, inadvertently reinforcing underlying inequalities [16,17,18,19,20,21,22,23]. As a consequence, LGBTI+ youth may have increased reluctance to disclose mental health difficulties due to concerns that providers may misunderstand their LGBTI+ identity as the source of mental ill health, or lack understanding and awareness of appropriate language and terminology [24].

### 1.3. Social Justice as a Pre-Requisite for Wellbeing

The WHO makes a further contribution to understandings of wellbeing through the Ottawa Charter for Health Promotion, which emphasises that social justice is a necessary pre-requisite for health, including mental health [25]. While there are multiple definitions of justice, distinctions have been drawn between redistributive and recognitive forms of justice [26]. Honneth concurs, highlighting the importance of recognition and revaluing disrespected identities through promoting cultural diversity and group differentiation [27,28,29]. In particular, Honneth underscores the importance of recognitive justice for emancipation struggles, using this as an example of social justice for LGBTI+ communities, described as: “culturally integrated communities with a common history, language and sensibility” [29] (p. 162). His tripartite framework emphasises three interconnected forms of recognition: interpersonal, community and legal relations [27,28,29]. This extends recognition beyond intersubjective relationships to the structural context through the recognition of universal human rights [27,28,29].

### 1.4. Rationale and Objectives

From our initial analysis and literature research, there is no existing review (narrative, systematic or scoping) on protective factors for LGBTI+ youth wellbeing. The limited research focus on protective factors is noteworthy given that almost two decades have passed since Meyer drew attention to the potential of “stress-ameliorating factors” for mental health [13] (p. 678). Further, a decade ago, Haas et al. specifically recommended that studies should be conducted on potentially protective factors for LGBTI+ populations [4]. The methodological framework for scoping reviews was followed as outlined by Arksey and O’Malley and Levac, Colquhoun and O’Brien [30,31]. The review aimed to: map the concepts, themes and types of available evidence within the existing literature; describe the characteristics of those studies undertaken to date, the various domains assessed and the specific outcome measures used; and to identify research deficits and knowledge gaps [30,31,32,33]. In accordance with the PRISMA Extension for Scoping Reviews (PRISMA-ScR) reporting statement [34], outlined in Appendix B, a protocol was published a priori (https://hrbopenresearch.org/articles/3-11, accessed on 28 October 2021) [35]. Supplementary Data (see Appendix A) provided with this review, were submitted to the Open Science Framework repository [36]. The overarching objective was to collect and synthesise evidence on the protective factors for LGBTI+ youth wellbeing.

This scoping review has potential to inform policy, practice and research, particularly through mapping a course forward to guide the planning and the commissioning of future studies [30,31,32,33].

## 2. Materials and Methods

### 2.1. Search Strategy

The search strategy was developed by the lead author (N.C.) in consultation with a subject liaison librarian (M.B.), with detailed search terms subsequently generated (N.C. and L.T.). No restrictions on time were placed on the search. With the limited research attention given to the topic, the review focused on published academic, peer-reviewed research articles and review articles in English. The Population–Concept–Context (PCC) approach informed inclusion screening criteria [34]. This was conceptualised as: P—Population: sexual minority, gender minority, intersex and non-binary youth; C—Concept: protective factors that promote wellbeing; and C—Context: any country, with broadly comparable supportive environments, as outlined in the Global Acceptance Index (GAI) [37]. Study selection was based on a priori eligibility criteria as outlined in Table 1.

### 2.2. Search and Study Selection

Studies were identified through electronic academic database searches using a combination of title and keyword terms alongside MeSH headings across six databases: PubMed; CINAHL; PsycINFO; ASSIA, Eric ProQuest; and Academic Search Complete. Prior to searching, the full electronic search strategy for PubMed was deposited in the Open Science Framework repository [36]. Comprehensive searches were conducted across all databases on 21 June 2020 and citations were managed using the bibliographic software manager, EndNote, with duplicates removed and imported into Covidence (N.C.).

All titles and abstracts were screened by two reviewers independently, in two teams (N.C. and O.J.; N.C. and L.T.), using screening tools tested by the team before their use (N.C., O.J., L.T., M.B. and D.C.) [38,39,40]. Disagreements were resolved via discussion, with reference to the a priori eligibility criteria until consensus was achieved (N.C., O.J., L.T. and D.C.). The PCC criteria were applied to the full text by two reviewers, independently (N.C. and O.J.) [34]. Another reviewer was recruited to assist in resolving disagreements (A.K.). CART criteria (Completeness, Accuracy, Relevance, Timeliness) were applied to intervention studies in relation to the research question (N.C. and A.K.) [41]. Following further discussion (N.C., A.K., O.J. and D.C.), records with medical, pharmacological and therapeutic interventions were excluded. While the authors acknowledge that treatment can promote wellbeing, the focus of this review is on protective factors that are health promoting [1,2,3,25]. This process sought to ensure robust, transparent decision-making informed by a clear rationale for selecting sources of evidence [38,39,40].

Follow-up search strategies included contact with 29 content experts, requesting information on relevant published studies, with a reminder sent one week later (N.C.). Over a third of authors responded (34.5%). This was complemented by reference checks for relevant publications and a final hand search of peer-reviewed journals by dissertation author name (N.C.). All identified records were cross-checked against Covidence and independently double-screened (N.C. and A.K.).

### 2.3. Data Charting and Summarising Results

Two study team members designed a template, to confirm relevance and extract characteristics from each full-text record (N.C. and D.C.). A pilot exercise was undertaken to guide the process, as recommended by Levac et al. (N.C., O.J., L.T., M.B. and D.C.) [31]. Based on this preliminary exercise, half of the records identified through database searching were extracted by a single reviewer (N.C.), with the study team completing checks against the original articles (A.K., O.J., L.T., M.B. and D.C.). The data items were compiled by the lead author (N.C.) in Microsoft Excel of the main details and relevant data collection variables (lead author, year of publication, study location, title, methodology and analysis, recruitment, demographic details, protective factors, wellbeing indicators).

### 2.4. Content and Thematic Analysis

Content and thematic analysis was undertaken, as per scoping review guidelines [30,31,32,33]. An inductive approach initially extracted protective factors, with a deductive approach subsequently applied across all records to assess the relevance of Honneth’s Recognition Theory [27,28,29]. The first author (N.C.) collated and categorised the records iteratively to summarise the results, with another member recruited to the study team to cross-reference charted data against the original articles (C.B.) [41]. Study team members regularly assessed this process to ensure consistency of the synthesis of results with the scoping review research question and purpose (A.K., L.T. and D.C.).

### 2.5. Consultation

Levac et al. recommend that the consultation stage is undertaken in order to enhance methodological rigour [31]. Further, Daudt et al. suggest that suitable stakeholders should be invited to be part of the research team [33]. The study team included members from within LGBTI+ communities with research, policy and practice backgrounds. Ethical approval was granted from a university Humanities Research Ethics Committee to undertake an online stakeholder consultation complemented by online discussions with LGBTI+ young people and peer allies (HS-19-80) [42,43,44,45,46]. Using an iterative Consulting–Conducting–Collaborating–Checking cycle for “learning *with*” LGBTI+ youth and allies, young people were invited to share their thoughts and insights [47,48]. The overarching process for obtaining and confirming data was underpinned by the work of Pollock et al. to ensure correct data interpretation and suggestions for knowledge translation [49]. This process enhanced the data analysis.

## 3. Results

### 3.1. Search Results

Following deduplication, 2902 records were double-screened, with 132 additional records located via content experts, reference lists and hand searches. All sources of evidence were screened, duplicates removed, and those published after the date of the search excluded. In total, 58 records were identified through database searches, with a further 38 additional records meeting eligibility criteria. This iterative screening and filtering process, with reasons for exclusion recorded at each stage, is illustrated in the flow diagram in Figure 1.

### 3.2. Overview of Documented Records

The review identified 96 records spanning just over three decades, from 1989 until 2020. All 96 records are presented in Table 2, Table 3, Table 4, Table 5, Table 6 and Table 7. While the first identified records date from 1989, it is notable that it was a further ten years before there was an exponential increase in research attention on, or including, factors that protect or promote LGBTI+ youth wellbeing. As such, the first 20 years of this review account for just 10.3% of records, with 89.7% of records published since 2010. An overview of these findings is illustrated in Figure 2.

While 25 countries met context inclusion criteria [37], only the United States (*n* = 79), Canada (*n* = 11), Australia (*n* = 4), Britain (*n* = 3) and New Zealand (*n* = 2) were represented. This is consistent with a recent landscape review and research gap analysis identifying the paucity of research, across Europe, of any persuasion, focused on LGBTI+ youth [50]. From the searches, we reviewed quantitative (*n* = 45), qualitative (*n* = 34) and mixed-methods research (*n* = 8) studies, with sample sizes ranging from *n* = 5 through to *n* = 4314. Systematic (*n* = 5) and narrative (*n* = 4) reviews were also included (Figure 3). Four of these provide a global perspective. Quantitative research accounts for almost half of the records (46.9%). It is notable that it was not until 2014 that these records included research using large, population-based datasets, with variables on sexual orientation, measuring identification, rather than attraction or behaviour. The emergence of population-based analyses in relation to gender identity is more recent, dating from 2018. Prior to this, studies recruited participants mainly through LGBTI+ organisations, community venues and events.

### 3.3. Demographic Overview

#### 3.3.1. Orientations and Identities

A fundamental issue across all included sources related to definitions, terminology and self-descriptors used by research participants. Research was predominantly conducted with sexual minority youth populations (72.9%), with studies including transgender and gender minority youth being more sparse (25.9%) (Figure 4). There is a paucity of research with intersex youth and those with variations in sex development (1.2%). One study focused on those who identify with a medical term: congenital adrenal hyperplasia (CAH).

#### 3.3.2. Self-Descriptors

There is broad variation in the range of identities and orientations included. Alongside lesbian, gay, bisexual, transgender and intersex from the LGBTI+ acronym, as Figure 3 highlights, the “+” symbol encompasses 57 forms of self-identification: 23 self-descriptors used by sexual minority youth; 17 self-descriptors for those who are transgender; with 16 terms encompassing gender minority identities; and one using medical terminology for variations of sex development.

#### 3.3.3. Being “Out”

In relation to identity and orientation, some records highlight that openness regarding sexual orientation and/or gender diversity is critical for positive wellbeing [51,52,53] and is associated with reduced stigma and discrimination [51,52] and increased pride [54]. Sexual minority youth “out” to a larger peer network reported higher levels of support, particularly with longer lapses of time since disclosure [55]. However, both concealment and its opposite, open disclosure of sexual orientation and gender identity, may be equally protective [52,56]. These nuanced findings emphasise that it may depend on who young people disclose to, with both youth who were fully “out” and those not “out” at all having to manage these dynamics least [52,57]. Further, the motivation to conceal may not negatively impact on connectedness, including within LGBTI+ networks [58].

#### 3.3.4. Age

The records included LGBTI+ populations aged between 10 and 24 years, in accordance with the definition of youth [59,60]. Where reported, the mean age, across primary studies, ranged from 14.3 years to 23.4 years. As illustrated in Figure 5, most of the research focus has been with emerging adults, aged over 18, with some focus on adolescents aged 14–17 years. One study that included teenagers noted that there were far fewer younger participants [61]. There was limited research attention on children aged 10–13 years.

The combined inductive content analysis and deductive thematic analysis found several areas of interest, highlighting the interpersonal, community and legal factors associated with LGBTI+ youth wellbeing (Figure 6). Key themes included: intersubjective recognition; community connectedness; inclusion through universal rights; and intersecting forms of recognition. These protective factors will now be discussed, before outlining the impact on outcomes and associations with broad indicators of wellbeing.

### 3.4. Intersubjective Recognition

Honneth highlights that recognition through interpersonal relationships with an other whom one mutually recognises supports the development of security and resilience, with an impact on self-confidence [28] (pp. 26–29). Intersubjective relations included relationships with parents (*n* = 40), peers (*n* = 32) and providers (*n* = 22), with the proportion of records illustrated in Figure 6. Interpersonal protective factors and the impact on wellbeing are tabulated, in chronological order from the most recent, in Table 2 of the quantitative records (*n* = 16), Table 3 of the qualitative and other records (*n* = 9), alongside those captured in Table 7 of the intersecting records (*n* = 15).

#### 3.4.1. Parents

Families play a vital role in LGBTI+ youth wellbeing, with relationships showing the greatest promotive effects. Parental acceptance and affirmation, belonging and connectedness, understanding and advocacy were all associated with increased wellbeing for LGBTI+ youth. Belonging and connectedness [56,61,62,63,64,65,66,67,68,69,70,71,72,73,74,75,76,77] and enhanced emotional support and closeness [55,63,65,75,76,77,78,79,80,81,82] included positive experiences, comments, behaviours and interventions [74,83]. Positive attitudes of parents extended to self-education and seeking to understand their child’s sexual orientation [71,74,75,76,80,84], gender identity [69,74,83,85] and sex development [86]. Advocacy was also highlighted and included support for, and assistance with, accessing care [67,79,87]. Accepting and affirming parental attitudes [65,68,69,70,71,72,73,74,75,76,77,78,80,81,82,83,84,85,88,89,90] were pivotal in facilitating identity exploration [65,69,70,73,75,80,85,90]. However, there is nuance in these findings. Some records note that acceptance and affirmation may be more likely to come from families rich in other forms of support [88,91]. Further, parents may be less likely to provide LGBTI+ identity-specific support [55,89]. While more generalised forms of support are valued, this may require negotiation by young people [51,55,56,89]. Family support was particularly important for younger sexual minority youth, with the positive effects of peer support increasing with age [77,88,92,93]. This highlights the potential promotive effects of parental support for independence and autonomy during emerging adulthood [80].

#### 3.4.2. Peers

Social support from peers was identified as a protective factor in the lives of LGBTI+ youth (*n* = 32). A range of peer relationships were identified: close friendships with gender and sexual minority peers [52,53,55,61,62,68,69,88,89,90,91,92,93,94,95], including romantic relationships [53,90,96,97,98,99], cross-sexual orientation friendships [52,55,100,101,102] and supportive peer relationships [52,53,57,61,65,69,72,75,88,90,93,94,95,102]. Such relationships with peers were regarded as mutually beneficial and reciprocal and of increasing importance as young people became older [72,77,88,92]. Differences were noted, with sexual minority friends described as providing support for coping with both sexuality stress and other problems, while heterosexual friendships provided more non-sexuality support than sexuality-related support [55,89,94]. While being “out” was associated with larger networks and a greater proportion of extremely close friends, it was also associated with a greater loss of heterosexual friendships after coming out, with fears regarding romantic relationships as a consequence [53]. As such, there may be limitations to the emotional support provided by heterosexual peers [102]. However, the promotive benefits of cross-sexual orientation friendships offer the potential for the appreciation of commonalities, breaking down negative stereotypes and increasing the sensitivity of the heterosexual friend to sexual minority perspectives [52,55,95,100,102].

#### 3.4.3. Providers

Providers and non-parent adults were also found to have a protective role (*n* = 22). This included those in formal and informal roles within education, youth work, health and social care or counselling and therapeutic roles. Such relationships provided opportunities for connectedness [62,66,81,90,101,103], belonging [66,104], support [77,83,90,105,106,107,108] and acceptance [70,109]. Particular skills were noted, including provider knowledge [70,75], provision of affirming care [86,110] and informal mentoring [73,98,110,111,112].

### 3.5. Community Connectedness

Honneth equally emphasises the unique contribution of community members, with the acknowledgment of individual contributions to the collective, enhancing self-worth (p. 30) [28]. Such protective community relations extended to LGBTI+ networks (*n* = 32), online connectedness (*n* = 10), faith communities (*n* = 10) and cultural communities (*n* = 5). Community protective factors and the impact on wellbeing are tabulated, in chronological order, in Table 4 of the quantitative and mixed-methods research, alongside the systematic review records (*n* = 12), Table 5 of the qualitative records (*n* = 17) and the records of intersecting forms of recognition in Table 7 (*n* = 19).

#### 3.5.1. LGBTI+ Communities

The theme of protective LGBTI+ communities was consistent across the records, with connection to LGBTI+ communities [54,65,67,68,70,72,73,75,90,93,95,100,101,102,113,114,115,116,117,118,119], alongside specific mention of gay [93,95,117,118], lesbian [54,58,100] and trans [83,104,106,107,110,120,121,122] communities. However, bisexual youth do not appear to derive such promotive benefits. This is noteworthy as research indicates that the largest proportion of those identifying as LGBTI+ are bisexual [6]. The records in the review attest to the importance of spaces and places, particularly LGBTI+ youth groups. Visibility of LGBTI+ communities was emphasised [98,100,106,108,112,114,115], achieved though LGBTI+ role models [73,75,93,100,101,117,123,124,125] and organisations [68,100,104,113,115], and through media representation [67,75,115,126].

#### 3.5.2. Online Communities

Online communities may provide important platforms for LGBTI+ youth, particularly those outside urban centres [65,73,103,124,126,127,128,129,130,131,132]. They appear to facilitate access to LGBTI+-specific social support [103,128,130,132], emotional support [132] and increased connectedness [73,124,126,129,131], as a consequence. This was enhanced when there were no in-person LGBTI+ supports available locally [131]. Access to online LGBTI+ communities provided a source of friendship and support, and offered the potential to find romance and to meet people in person [131,132]. Some online friendships may replace in person friendships [132]. Sexual minority youth were noted to be more adventurous in their online use, meeting people online, including for friendships and relationships, in contrast to their heterosexual peers [132].

#### 3.5.3. Faith Communities

This review identified potential for the presence of accepting faith communities to be a source of support for LGBTI+ youth [68,73,90,114,127,133,134,135,136,137]. While faith and LGBTI+ identities have often been assumed to be incompatible and mutually exclusive, those religions and communities with supportive attitudes, such as endorsing marriage equality, may enhance the interaction of diverse identities [127,135,137]. Positive acceptance may be promotive of identity development, which, in turn, mediates the relationship between identity integration and wellbeing [114,134,135,136].

#### 3.5.4. Cultural Communities

An emergent topic area identified as part of this review is the protective potential of cultural communities. Five records identify the potential for LGBTI+ identification and cultural identification to be mutually enhancing [70,73,103,106,125]. The interaction of these diverse identities provided support from peers and the inspiration of role models [70,106,125], resistance to cultural stigma and prejudice related to intersecting identities [103] and affirmation through the value of such unique and multifaceted identities [70,106]. Further, strong cultural and familial ties enhanced personalised coping strategies [125] and challenged the “at risk framing” of cultural messages [103].

### 3.6. Inclusion through Universal Rights

Honneth underscores that recognitive justice necessarily requires legal relations, i.e., recognition of universal rights and inclusion, which promotes empowerment and self-respect [28] (pp. 26–29). The scoping review highlights the importance of the structural context, beyond the broader social acceptance through legislative measures, captured by the GAI [37]. Documenting protective legal relations highlighted educational settings, in particular, with Gay–Straight Alliances (GSAs), also known as Gender–Sexuality Alliances, offering inclusive spaces. GSAs (*n* = 23), alongside inclusive policies (*n* = 23), curricular (*n* = 11) and extracurricular activities (*n* = 4) were all promotive of wellbeing. Such protective climates highlight the powerful protective potential of GSAs. These protective factors, and the impact on wellbeing, are tabulated in chronological order, from the most recent, in Table 6 of the legal relations and Table 7 of the intersecting protective factors.

#### 3.6.1. Gay–Straight Alliances/Gender–Sexuality Alliances (GSAs)

GSAs are student-run organisations that unite LGBTI+ young people and allies by providing support, opportunities to socialise, information and access to resources. This review found that the presence, alone, of GSAs in schools was protective [87,91,101,105,108,111,122,138,139,140,141,142,143]. This promotive benefit was enhanced through involvement and participation [64,87,105,111,139,140,142]. Further, greater levels of engagement were associated with greater benefits [64]. Even moderate levels of peer and significant other support appear to play a protective role [91]. GSAs may also facilitate access to LGBTI+ community networks [64,68,75,83,90,101,104,120,140,142,143].

#### 3.6.2. Policies

A number of policy factors, particularly in the school context, that promote LGBTI+ wellbeing were documented. While there was some reference to anti-discrimination measures [75,110,141,144], this review noted the presence of inclusive policies, as a means to influence overall school climates. Such policies were applied universally across the school and extended to administrative measures to provide for chosen name and use of pronouns [106,107,109,110,120,144], inclusive bathroom access [75,144,145,146], alongside all-gender dress codes, such as uniforms [75,109].

#### 3.6.3. Curricular

This scoping review highlights the importance of a comprehensive and inclusive education curriculum [65,68,75,83,120,147,148]. While these findings largely relate to the secondary school context, an inclusive curriculum includes, but is not limited to, puberty, sexuality and relationship education [120,123,141,147,148]. In addition, curricula should have broader relevance to sexual minority youth, gender minority youth and youth with diverse sex development. As such, curricular education offers potential to extend beyond health, to ensure the representation of LGBTI+ lives throughout the humanities and sciences [68,70,75,93,111,123].

#### 3.6.4. Extracurricular Activities

Alongside inclusive spaces in schools for LGBTI+ youth, curricular provision may also co-exist alongside extracurricular activities, offered through school and outside educational contexts [65,120,147,148]. This included welcoming same-gender partners at school events, alongside the partners of staff and family from sexual and gender minority backgrounds [147]. There was specific mention of involvement in creative pursuits, such as music, art, dance and drama, alongside sports participation [120]. It is noteworthy that there was only one record that specifically mentioned creative and sporting extracurricular activities [120].

### 3.7. Intersecting Forms of Recognition

Honneth’s Recognition Theory outlines an intersecting, tripartite framework that underscores the co-existence and interconnection between interpersonal, community and legal forms of recognition. In particular, GSAs appear to offer powerful protective potential through the intersection of these forms of recognition. These promotive benefits are illustrated in Figure 7.

As such, GSAs may enhance allyship by peers and providers (*n* = 21) and facilitate access and connectedness to LGBTI+ community networks (*n* = 11). Positive affirmation of identities and orientations, and allyship by peers [64,68,73,83,90,100,110,123,140], alongside that of providers [64,67,68,73,75,83,90,100,101,104,110,123,140,141,142,143], may enhance and promote advocacy at both the individual and collective levels [64,69,70,83,85,105,106,107,109,110,120,123,138,140,141]. Advocacy, in turn, may promote activism, with strong associations with wellbeing [64,67,70,98,110,119,120,138,140,141,142,143]. Additionally, GSAs may facilitate and strengthen the development LGBTI+ community networks, enhancing community relations [64,68,75,83,90,101,104,120,140,142,143]. Further, the presence of GSAs was associated with the increased likelihood of co-existing inclusive policies (*n* = 12), inclusive school curricular (*n* = 5) and extracurricular activities (*n* = 1). These findings are tabulated, in chronological order from the most recent, in Table 7.

### 3.8. Indicators of Wellbeing

Of the 96 records included in this review, interpersonal relations, community connectedness, legal inclusion through universal rights and the intersecting tripartite forms of recognition were found to be associated with enhanced LGBTI+ youth wellbeing. This accords with Honneth’s Recognition Theory [27,28,29]. In particular, significantly better psychological outcomes were noted (*n* = 36). These included lower levels of depression [53,54,55,57,61,62,63,64,65,78,79,80,81,83,85,87,88,89,90,91,92,96,105,109,113,127,138,147], anxiety [55,64,78,88,91,105,113] and emotional or psychological distress [55,61,63,78,81,88,91,96,113,127,138].

All quantitative studies used self-report scales for depression, including the Center for Epidemiologic Studies Depression Scale (CES-D); the Beck Depression Inventory; the Diagnostic Interview Schedule for Children; Brief Symptom Inventory subscales for depression; the 2-item Patient Health Questionnaire-2; a single item from the WHO Composite International Diagnostic Interview Short Form; a combination of CES-D items with the Structured Clinical Interview for DSM-IV and Schedule for Affective Disorders and Schizophrenia for School-Age Children; and a question asking whether participants felt very “trapped, lonely, sad, blue, depressed, or hopeless about the future”. Internal consistency, measured by Cronbach’s alpha for the depression scales, where reported, ranged from .70 to .94. Some studies dichotomised scores to differentiate between depressive symptoms that were clinically significant. Qualitative studies garnered perceptions of self-reported, psychosocial consequences of supportive and unsupportive behaviours.

Measures for anxiety used the 21-item Beck Anxiety Inventory; the Brief Symptom Inventory subscales for anxiety; and a question asking whether participants were “anxious, nervous, tense, scared, panicked, or like something bad was going to happen”. Where reported, the coefficient alpha reliability estimate was α =.95.

Measures for psychological distress included the Brief Symptom Inventory; the Brief Hopelessness Scale; short form of the Global Appraisal of Individual Needs; the Emotional Symptoms Index of the Behavior Assessment System for Children; and the General Well-Being Schedule with a question measuring stress and despair. Where reported, the Cronbach’s alpha ranged from .80 to .94.

The amelioration of the negative effects of victimisation was also identified [57,62,63,66,87,88,91,96,138,139,140]. Measures included the Scope and Prevalence of Anti-Lesbian/Gay Victimization; family victimisation related to sexual orientation; bully victimization in the past 30 days; experience of violence at school in the past 30 days; a 10-item lifetime victimization on the basis of LGBT identity scale; an adapted scale of the California Healthy Kids Survey measure on violence, safety, harassment and bullying; a 10-item measure of the frequency of LGBT victimization; at-school victimization adapted from the Bullying and Victimization Scale; and experience of school victimization based on sexual orientation.

Decreased odds of non-suicidal self-injury were noted [61,62,66,67,93,109,113,120,133,134,141,147]. There were also reduced odds of suicidal thoughts, symptoms and attempts [62,63,65,66,67,78,81,87,91,93,105,109,120,133,134,139,141,147]. These positive impacts were associated with interpersonal, community and legal protective factors. For example, an increase by one context—be it interpersonal, within the community or enshrined in policy—supporting chosen name use, predicted a decrease in depressive symptoms, suicidal ideation and suicidal behaviour [109]. Further, disparities in suicidal thoughts were nearly eliminated in US states with the most protective school climates [141].

Measures of suicidality included questionnaire items on self-harm or self-injury that was non-suicidal in intent (NSSI). This was dichotomised regarding frequency and/or recency. Experience of suicidal ideation and attempt in the past year was also measured, with a single-response question and indicator of frequency.

It is critically important that over a third of records (37.5%) noted such reductions, given the concern at the higher prevalence of psychological distress and suicidality for LGBTI+ youth populations [9,10,11,12,15]. This underscores the resonance of Meyer’s call for research attention on “stress-ameliorating factors” [13] (p. 678). This also accords with Honneth, who emphasised that recognition extends beyond the interpersonal and community level, highlighting that recognitive justice exists within broader structural contexts [27,28,29]. A broad range of wellbeing indicators, associated with holistic forms of recognition, were mapped onto Honneth’s tripartite framework, as illustrated in Figure 8. This is consistent with the WHO constitution, which notes that health is more than the absence of disease [1].

### 3.9. Consultations

There was broad consensus of these findings through the stakeholder consultation, complemented by online discussions with LGBTI+ youth and peer allies. Presentations of the preliminary findings were followed by dialogue and feedback [43]. Stakeholder discussions were guided by the policy-makers and researchers in attendance, and focused on the challenges in capturing the breadth of diversity within identities and orientations, especially for quantitative studies, with particular reference to appropriate question wording for the inclusion of non-binary and intersex youth. The LGBTI+ acronym has particular resonance in the Irish context, with the inclusion of intersex evolving iteratively through research and policy-making processes [24,149]. A more comprehensive qualitative study is being conducted with LGBTI+ youth, living in Ireland, and includes consultation on the phrasing and placement of demographic questions, with the potential to influence future waves of longitudinal data collection. Findings from this research will be published in a follow-up manuscript.

During consultations with LGBTI+ youth and peer allies, the critical role of interpersonal relations with parents, peers and providers was reiterated. In particular, affirming and accepting behaviour (especially from family) was recognised as extremely protective. Young people confirmed that broader LGBTI+ communities and, especially, connectedness to gay, lesbian and transgender communities, play an important role, including as chosen families. The young people were initially surprised by the potential of faith communities to be protective. With further discussion, they suggested that such communities may be supportive of LGBTI+ identities because of, rather than despite, their faith. Within the study team and with the stakeholder and youth consultations, intergenerational differences were noted in relation to online communities. While the full study team were aware of potential harm from online activity, including cyberbullying [150], younger team members and consultations with youth concurred with the description of online communities as a “safety net” [132]. Young people also understood the broad lack of awareness of this promotive impact—for example, the potential of having an avatar online that appropriately reflects a young person’s self-expression [129]. This is, perhaps, reflected in the differences reported between sexual minority youth and their heterosexual peers [132]. The young people commented that for transgender and gender minority youth, in particular, such online communities are “literally lifesaving,” due to geography and population size.

Stakeholder and youth consultations confirmed the importance of GSAs. This reflects the nationwide youth consultation conducted for the Irish LGBTI+ National Youth Strategy, with young people calling for the introduction of such alliances in Irish schools [149]. This attests to the idea of “learning *with*” LGBTI+ youth [47], and suggests that young people are engaged and aware of what is happening for LGBTI+ communities globally.

## 4. Discussion

Holistic, and deliberately broad, conceptualisations of wellbeing, underpinned by the WHO, and complemented by Honneth’s Recognition Theory, informed this scoping review. The findings underscore the nuance and breadth of factors that may potentially promote LGBTI+ youth wellbeing. The critical importance of family and friends is highlighted, including LGBTI+ chosen families, and extending to online networks. Community connectedness with faith and cultural communities emphasises the need to acknowledge that young LGBTI+ lives are intersectional, with multi-faceted, diverse identities. Protective school climates that are inclusive appear to have an important promotive role. This review notes the powerful, protective potential of GSAs. The creation of such safe spaces may be particularly important for youth who are exploring their orientations and identities, offering the potential for peer and provider allyship.

The size and breadth of the records included in this review indicates an exponential increase in research attention on this topic, particularly in the past decade. This, perhaps, reflects the call by Hass et al. for an increased focus on protective factors [4]. However, it is in stark contrast to the extensive research focus on psychological distress and suicidality [10,11,12]. There is a pressing need for increased research attention on protective factors that promote LGBTI+ youth wellbeing. The more recent availability of population-based datasets that facilitated representative and generalisable analyses is welcome, and the prioritisation of secondary analysis and further comparative research is recommended. While not comparative, the quantitative records included in this review capture a wealth of experiences, with a continued need for such research. The rich nuance of the qualitative studies emphasises the need for an increased focus on these methods, while the paucity of mixed-methods research calls for greater investment in these methodologies. Convenience, purposeful and snowball sampling via LGBTI+ organisations, community venues and events seems appropriate, given the importance of such communities. This could be further enhanced by increased attention to alternative forms of recruitment [95].

The authors call for continued research with sexual minority youth, an increased research emphasis with transgender and gender minority young people and the urgent prioritisation of research attention for youth with variations in sex development [9,15]. However, it is recognised that the inclusion of intersex, within the broader LGBTI+ acronym, continues to generate discussion. For example, the recent Australian human rights commission report, while using this acronym, noted that the needs and context for intersex youth are unique, and are not encompassed by terms related to sexual and gender minority populations [151]. This accords with the recent work of the National Academies [9]. It is acknowledged that the challenge posed for practitioners, policy-makers and researchers is not insubstantial in relation to the call for continued and increased attention on sexual and gender minority youth, and the pressing need for the prioritisation of focus on populations with diverse sex development. However, this offers rich opportunities to explore the breadth and depth of LGBTI+ youth’s lived experience. This is now discussed in relation to the nuance in these findings regarding multi-faceted orientations and identities; broadening understandings of family; the salience of community connectedness; shifts from protectionism to rights-based, universal inclusion; and mental health beyond a dichotomy.

### 4.1. Multi-Faceted Orientations and Identities

Social acceptance and increased visibility may facilitate broader understandings of sexual orientation, gender identity and sex development [37]. The authors call for increased attention to disaggregating data on sexual orientation, with particular attention given to bisexual youth, due to prevalence [6,7], alongside concerns that the protective factors identified in this review may not have the same promotive benefits. As Figure 3 highlights, young people perceive sexual orientation and gender identity as dynamic and are comfortable and confident with a myriad of forms of self-identification. It appears that a proportion of young people no longer assume rigid sexual orientation labels and binary gender identities [152]. While this raises challenges for researchers in relation to measurement [9], it offers opportunities for “learning *with*” LGBTI+ youth, alongside those who identify beyond this acronym, and their peer allies [47,48]. This reinforces the need for preliminary, participatory research to understand appropriate self-identifiers as part of survey design and development. This necessarily extends to attending to possible non-medical self-descriptors for youth with diverse sex development [8,9].

Sparse research has included younger populations. In this regard, measuring attraction in relation to sexual orientation, rather than identification, may be of increased importance [6,7]. One study noted children’s early knowledge that they were not heterosexual, with an average age of 10.3 years for boys and 12.2 years for girls [53]. The Health Behaviour in School-Aged Children may provide an example of measuring attraction, with a pilot conducted across eight European countries [153]. While this offers potential, it poses additional challenges as measures of sexual orientation may assume a gender binary. An additional complexity, specific to LGBTI+ identities and inclusions, is the issue of parental consent for children and adolescent research participants [154]. However, some research ethics processes can accommodate passive parental consent, or waive this requirement [52,62,66,68,81,113,127,133,141,145].

### 4.2. Broadening Understandings of Family

The powerful protective role of family accords with research that identified the importance of “One Good Adult” [155,156]. This can be a family member, provider (in both formal and informal roles) or non-parent adult—someone who is available to the young person in times of need [155,156]. As such, caring adults within LGBTI+ communities may also form chosen family, alongside, or in lieu of, parents and adult family members [57]. In light of the pervasive and dominant focus on risk factors, it is, perhaps, understandable that supportive adults have concerns about LGBTI+ youth mental health distress. This may inadvertently lead some adults to seek to prevent young people from expressing their identities and orientations, in a mistaken belief that this may be protective [157]. It is recommended that a realist review, underpinned by the methodology outlined by Pawson and Tilley, be undertaken, predicated on complexity, rather than seeking to isolate social interactions [158]. This type of review could be contextualised within the work of organisations, such as the Family Acceptance Project, which, alongside demonstrating the benefits of affirming behaviours, offers insights into working with rejecting families and assisting them to support their children (see https://familyproject.sfsu.edu/, accessed on 28 October 2021).

Within broader understandings of family, although after the date that the search was run, an emergent topic area suggests companion animals, particularly family cats and dogs, can promote LGBTI+ youth wellbeing [159,160]. This may indicate the promotive effects of human–animal bonds [47]. The authors concur that this topic warrants further research.

### 4.3. Salience of Community Connectedness

This review highlights the importance of community, with a sense of connectedness via an LGBTI+ identity associated with collective self-esteem and positive self-identification [54,77,83,98,106,111,116,121]. This extends beyond Honneth’s concept of recognition of the individual contribution of community members [27,28,29], to acknowledgment of the importance of the wider contributions of LGBTI+ communities. As such, policy investment in LGBTI+ community endeavours and initiatives is of critical importance. In the current problem-focused funding climate, LGBTI+ community groups have been placed in an invidious position and may find themselves reinscribing a risk-based, deficit focus in order to maintain and secure further funding [23]. In particular, the benefits of involvement in LGBTI+ sporting, creative and social groups warrants further research attention, in light of the positive impact on wellbeing for adult members of LGBTI+ communities [161,162]. It is recommended that further investigation be undertaken to determine whether involvement in extracurricular activities through groups, by and for LGBTI+ communities, could be supportive for LGBTI+ youth. The authors extend this to online fora, and connectivity, via gaming and social networking.

The concept of a singular readymade “community”, which assumes an inevitable sense of belonging, is contested and the use of “communities” more appropriately reflects the diversity “within and between” those who identify as LGBTI+ [163]. The findings regarding faith communities and cultural communities counter the assumption of mutually exclusive identities. This has important implications and prompts practitioners, policy-makers and researchers to ensure that LGBTI+ youth’s lived experience is contextualised within intersectional understandings of the salience of identities that include sexual and gender minority orientations and identifications, alongside faith, ethnicity and cultural diversity [164]. The nuance regarding the potential promotive benefit of religious belief and spirituality is captured in the systematic review by McCann et al. [135]. Understanding of these contexts may be enhanced with reference to institutional allyship, beyond interpersonal allyship by members of faith communities, to those embedding institutional allyship, predicated on values of social justice, equity, diversity and inclusion [165]. This review calls for a greater focus on the promotive benefits of ethnic and cultural communities for LGBTI+ identified young people. This accords with recent research highlighting the importance of community belonging for Black LGBTQ adult mental health and wellbeing [166]. Further research may be strengthened with reference to Indigenous peoples’ understandings of the fluidity and blurring of sexual minority and gender minority identities beyond the LGBTI+ acronym [167,168].

### 4.4. Shifts from Protectionisism to Rights-Based, Universal Inclusion

Much of this review has focused on the educational context, in light of the ages encompassed by the term “youth”, which encompasses those aged 10–24 years and therefore likely to experience primary and secondary education, and possibly higher education contexts [59]. The authors call for greater research into all aspects of legal relations. Protective school climates appear to be critically important, beyond a focus on protectionist approaches, which inform anti-discrimination measures and seek to address bullying [16,17,18,19,20,21,22,23]. Rather, this review highlights the potential benefit of strengths-based policy measures of provision for all students for chosen name and pronoun use [109]; inclusive access to all-gender bathrooms and changing rooms [169]; and inclusive dress codes, such as all-gender uniforms [75,109]. This extends to policy and curricula, with recommendations for puberty, sexuality and relationship education [170], inclusive education [171] and embedding an ethos of diversity and inclusion within schools, with potential promotive benefits for all [172,173,174].

The powerful, protective potential of GSAs is noted. The creation of such safe spaces may be equally important for youth who are “out”, those exploring their orientations and identities and those who do not disclose. This accords with findings that being completely “out” or completely “in” may be protective [51,52,53,54]. The design of GSAs, with allyship central to these alliances, appears to facilitate participation without young people being required to declare their identities or orientations. GSAs may potentially provide an inclusive space to challenge rigid, binary conceptualisations of gender and sexuality, explore ambiguities and ensure the visibility of a diverse expression of identities and orientations [152]. This may foster a sense of connectedness and school belonging. While inclusive provision may seek to address the needs of LGBTI+ self-identified students, the benefits appear to be far-reaching, with a suggestion of potential promotive benefit to all. The authors suggest that a realist review is undertaken to determine what works, for whom, in which contexts, in relation to the impact of GSAs, and resultant policies of inclusion, across multi-faceted, intersecting identities, including sexual orientation, gender identity, faith, ethnicity, socio-economic status and ability. While such alliances and policies are predicated on rights-based, inclusive provision that is universally available to all students, it is important to establish how this is extended to youth with multiple marginalised identities. Further, peers and providers are uniquely positioned to advocate for the importance of inclusive policies, including the provision of GSAs. As role models, informal mentors and allies, through advocacy and activism, may foster optimism and instil hope for the future, including future possible selves. Such allyship, at the interpersonal, intergenerational and institutional level, is associated with promoting LGBTI+ youth engagement, involvement and participation [149,152,175]. A systematic review on interpersonal, intergenerational and institutional allyship, provided by peers and providers, within the policy context may yield important insights [165,175,176].

### 4.5. Mental Health beyond a Dichotomy

The concerns regarding mental health disparities for LGBTI+ youth are well established [4,10,11,12]. While it is essential that the immediate and lasting factors that negatively impact on LGBTI+ youth wellbeing are not diminished or underrepresented, it is perhaps understandable that research attention has focused on mental health disparities [13]. However, this review identifies potentially “stress-ameliorating factors” [13] (p. 678), with interpersonal, community and legal factors associated with reductions in psychological distress and suicidality, alongside increased wellbeing. As such, experiences of mental ill health do not preclude experiences of mental wellness. Equally, it is important that the concept of resilience is not suggested as a solution to experiences of prejudice, discrimination and victimisation, exacerbating mental health stigma as a consequence [16]. This underscores the importance of strength-based approaches, predicated on nuanced conceptualisations of mental health beyond a binary of illness and wellness as dichotomised and mutually exclusive [1,2,3]. This has implications for policy, practice and research, beyond deficit-informed and protectionist approaches. In turn, needs assessments can explore strengths within young people’s lives, providing a basis for determining the protective potential of intersubjective, community and legal factors, those which can be enhanced, alongside factors requiring additional scaffolding. Such approaches recognise youth social and cultural capital and may connect young people to their own sense of competence and agency [47]. The authors call for a greater emphasis on broader conceptualisations of LGBTI+ youth wellbeing and recommend that equal priority is given to research on protective factors.

### 4.6. Limitations

The authors are heartened by the exponential increase in research focused on, or including, factors that protect or promote LGBTI+ youth wellbeing, particularly within the last decade. As Figure 2 illustrates, the number of records doubled in 2010 and again in 2016, with this trend also reflected in publications from 2020. The Figure 1 flowchart captures the many recent, relevant studies forwarded by context experts, outside the date that the search was run and not included in this review. Further, as this review focused on peer-reviewed, published, academic literature in English, it is possible that some records may not have been identified, particularly if studies were not indexed at the time of search, or used terms not included in the search string. While the review intended to include dissertations, due to embargo and repository restrictions, these could not be retrieved. This highlights the critical importance of publication that enhances the more rapid dissemination of research in a field where gaps in the literature remain pervasive.

This review focused specifically on self-identification in relation to sexual orientation, with studies including measures of attraction and recoded for identification excluded from this review. We further acknowledge that the use of terms relating to resilience is both limited and limiting, particularly in light of the experience of victimisation and stigmatisation for LGBTI+ youth. These findings may have been further enhanced by attention to the wealth of grey literature, including books, book chapters and reports. The authors recommend further scoping of this literature, particularly as it appears that policy-makers and practitioners may have already adopted strengths-based approaches. This highlights the importance of practitioner-informed research as an essential component of a virtuous research cycle. No studies on interventions were included and the authors recommend that a systematic review is conducted of educational, community-based, psycho-social, psychological, pharmacological and surgical interventions.

While the concept of recognitive justice remains contested, particularly in light of the importance of redistributive forms of justice [29], Honneth’s Recognition Theory provides a useful framework for scoping such tripartite, protective factors [26,27,28], with their interconnection illustrated in Figure 5. This also underscores the importance of attending to intersectionality, particularly that of LGBTI+ orientations and identities alongside socio-economic status [164].

Despite these limitations, this scoping review provides a nuanced, comprehensive overview of this body of literature.

## 5. Conclusions

The findings contained in this scoping review demonstrate that, rather than an LGBTI+ identity being assumed as a proxy for risk, there is a pressing need to attend to specific psychosocial strengths rather than the predominant focus on stressors for this population. The de-pathologising of LGBTI+ identities may be reflective of increased recognition, beyond the interpersonal and community level. Bringing a social justice perspective to this review, underpinned by Honneth’s Recognition Theory, is of critical importance, given the broad consensus of the elevated risk of psychological distress, self-harm and suicidality for LGBTI+ youth populations. It is with some urgency that this review concludes with an appeal for research funders and policy-makers to move beyond the dominant discourse focused solely on LGBTI+ youth’s mental health risk, which subsequently informs protectionist approaches. These findings have important practice and policy implications, highlighting the broad applicability of strengths-based approaches in assessment and the crucial need to develop mechanisms, underpinned by recognitive justice, to herald a change in the funding of future research directions. This emphasises the salience of enhanced understandings of inclusion, which is rights-based, universally available and of potential benefit to all.

## Figures and Tables

**Figure 1 ijerph-18-11682-f001:**
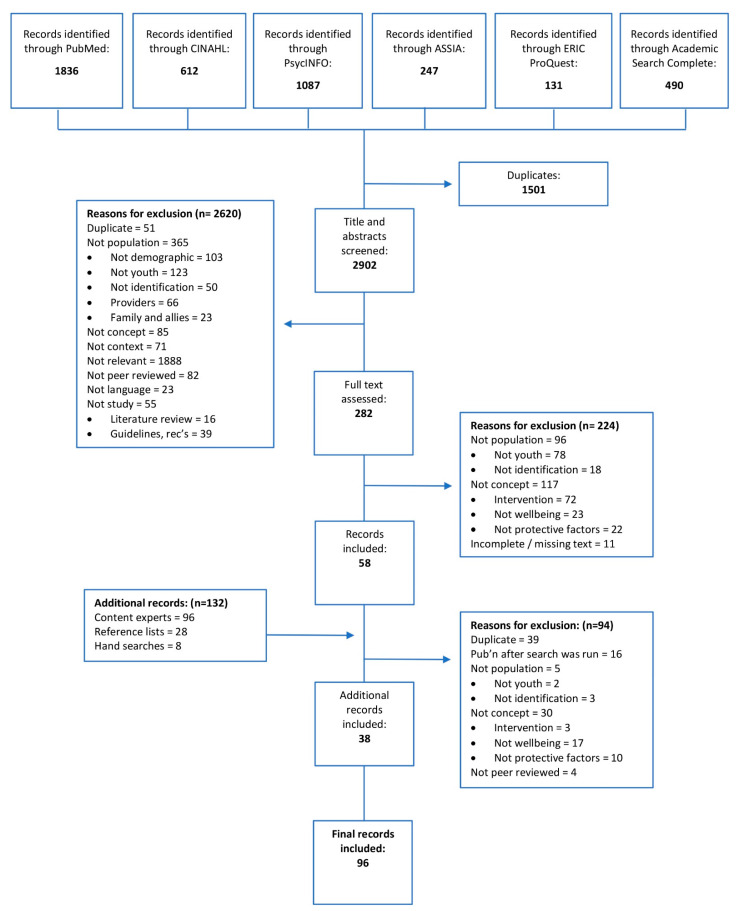
Screening and filtering process.

**Figure 2 ijerph-18-11682-f002:**
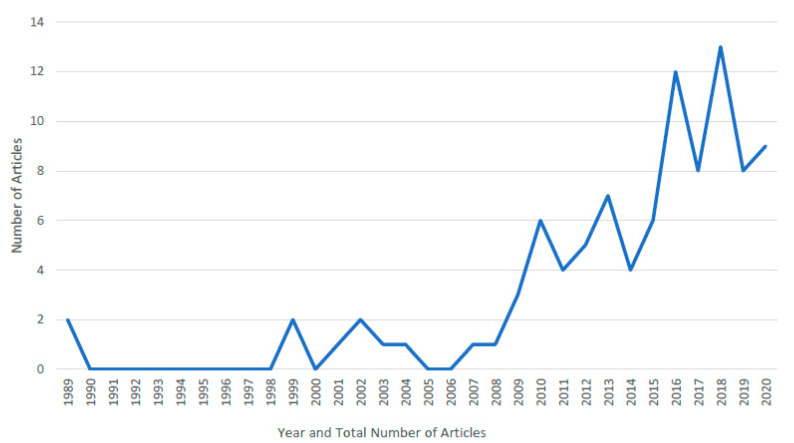
Research records achieving the inclusion criteria 1989 to 2020.

**Figure 3 ijerph-18-11682-f003:**
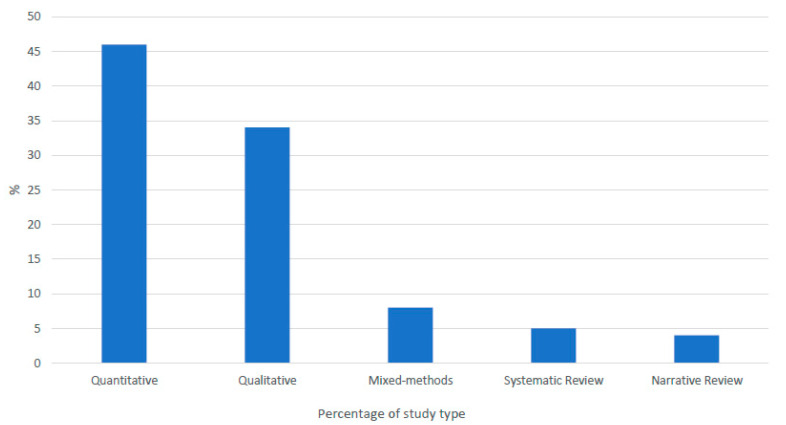
Research methodology for included records.

**Figure 4 ijerph-18-11682-f004:**
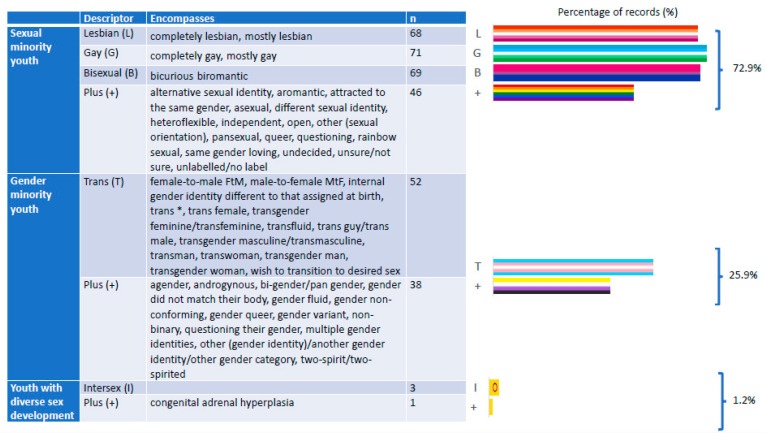
Self-descriptors and proportion of records focused on sexual minority youth, gender minority youth and those with diverse sex development.

**Figure 5 ijerph-18-11682-f005:**
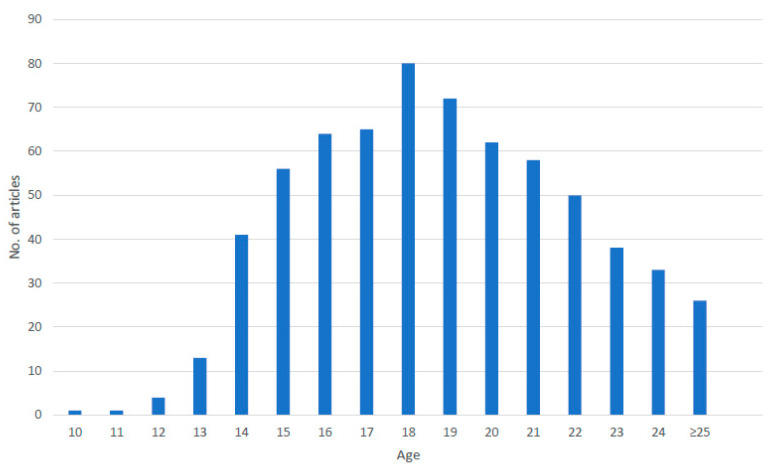
Participant ages across included records.

**Figure 6 ijerph-18-11682-f006:**
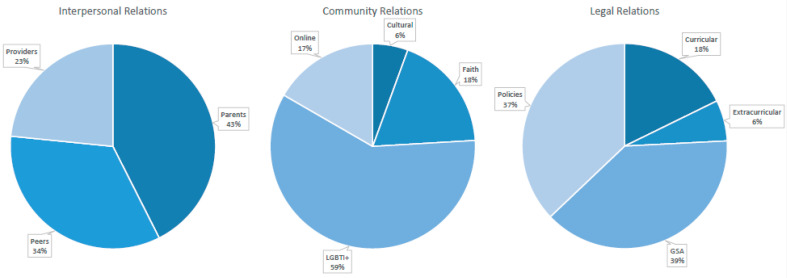
The proportion of interpersonal, community and legal factors across included records.

**Figure 7 ijerph-18-11682-f007:**
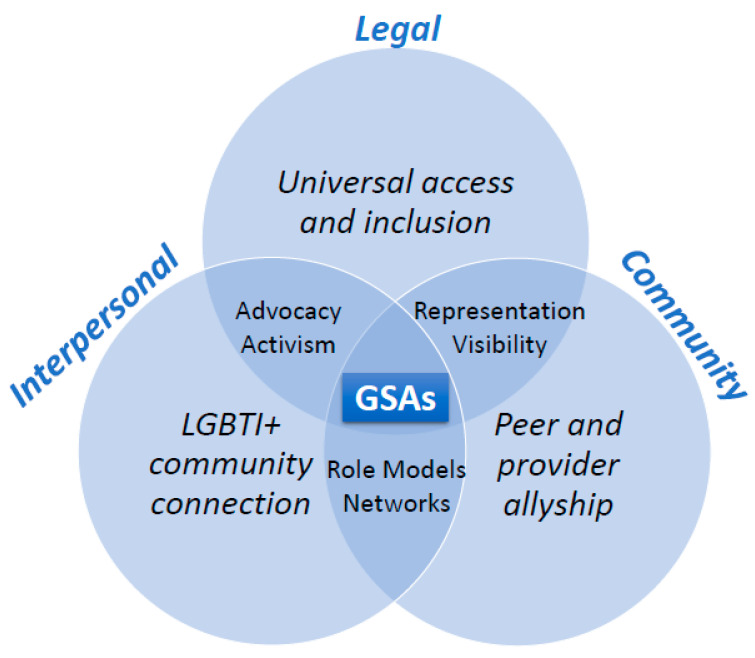
The association of Gay-Straight Alliances/Gender-Sexuality Alliances (GSAs) with intersecting interpersonal, community and legal protective factors.

**Figure 8 ijerph-18-11682-f008:**
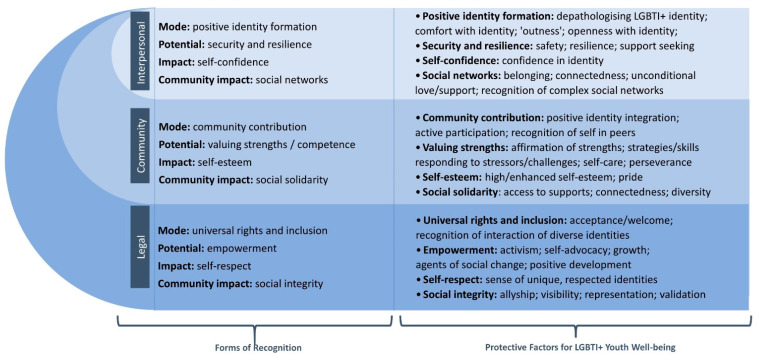
Mapping protective factors for LGBTI+ youth wellbeing onto Honneth’s Recognition Theory.

**Table 1 ijerph-18-11682-t001:** Inclusion and exclusion criteria of study selection using PCC criteria.

PCC	Inclusion	Exclusion
P—Population	Study includes participants who self-identify as lesbian, gay, bisexual, transgender, intersex, queer, questioning, asexual, non-binary or related terms Study with participants aged 10–24 years Study where young people are specifically targeted Study whereby the mean age falls within the specified age range	Heterosexual and/or cisgender participants only No demographic measure of sexual orientation, gender identity or non-binary or intersex status Study whereby participants are children ≤ 10 years or adults ≥ 24 years Study whereby the mean age falls outside the specified age range
C—Concept	Study referring to any measures of resilience Study referring to ecological, psychosocial or cognitive measures that protect wellbeing Study referring to “stress-ameliorating factors”	No reference in study to resilience No reference to any protective factors including: interpersonal, community-based or policy measures No reference to factors that mitigate minority stress
C—Context	Study conducted in a country (or region) with a broadly similar Global Acceptance Index rank	Study conducted in a country (or region) with a widely disparate Global Acceptance Index rank

**Table 2 ijerph-18-11682-t002:** Quantitative records of interpersonal relations: parental, peer and provider protective factors for LGBTI+ youth wellbeing (*n* = 21).

Author/Year/Location Title	Methodology/Analysis	Demographic Details	Protective Factors	Wellbeing Indicator
Parra et al., 2018 Canada The Buffering Effect of Peer Support on the Links Between Family Rejection and Psychosocial Adjustment in LGB Emerging Adults.	Quantitative In-person survey Measures included: sexual orientation disclosure, family attitudes, peer social support, anxiety/depressive symptoms, internalised homonegativity, self-esteem	Participants (*n* = 62 youth) 17–27 years old (mean = 21.3) Self-identified as lesbian, gay and bisexual	Peer: Peer social support Availability of peers and helping behaviours by peer social network Mutually beneficial and reciprocal relationships Potential for peers to become families of choice	Positive peer relationships play vital roles in LGB wellbeing, feelings of acceptance, self-esteem Perceived peer support associated with less depression and internalised homonegativity Peer support moderated the link between negative family attitudes/anxiety and family victimization/depression
Whitton et al., 2018 USA Romantic Involvement: A Protective Factor for Psychological Health in Racially-Diverse Young Sexual Minorities.	Quantitative Part of a larger longitudinal merged-cohort study over five years with 8 waves In-person survey Measures at each wave included current relationship involvement, psychological distress, LGBT victimization	Participants (*n* = 248 youth) 16–20 years old at first wave (mean 17.9) Self-identified as gay, lesbian, bisexual, questioning or unsure	Peer: Involvement in a romantic relationship Committed partnerships, other than marriage Romantic relationships potentially protective in promoting psychological health	Beneficial for psychological health and reduced psychological distress Benefits from middle adolescence into adulthood Buffered negative effects of victimization Predicted lower psychological distress for Black and gay/lesbian youth However, potential risk factor for bisexual youth
Veale et al., 2017 Canada Enacted Stigma, Mental Health, and Protective Factors Among Transgender Youth in Canada	Quantitative Online survey Measures included: enacted stigma (self-injury, suicide, depression and anxiety) and protective factors (family connectedness, friend support, school connectedness)	Participants (*n* = 923 youth) Aged 14–25 years (mean 20.0) Fewer (*n* = 323) 14–18 age group than 19–25 (*n* = 600) Self-identified as trans, genderqueer or felt that their gender did not match their body	Parent: Family connectedness Peer Friends caring Legal: School connectedness, e.g., through Gay–Straight Alliances/Gender Sexuality Alliances	Parental, peer and school support associated with favourable mental health outcomes/lower levels of enacted stigma Family connectedness strongest protective predictor against mental health difficulties School connectedness significant protective factor for extreme stress/despair
McConnell et al., 2016 USA Families Matter: Social Support and Mental Health Trajectories Among Lesbian, Gay, Bisexual, and Transgender Youth	Quantitative Part of a larger ongoing longitudinal study of LGBT youth In-person survey Measures used: Lifetime LGBT victimization, social support, mental health outcomes	Participants (*n* = 232 youth) 16–20 years old (mean 18.8) Self-identified as lesbian, gay bisexual, transgender, queer, questioning, attracted to the same gender	Parent/peer Family support and support from friends/peers and significant others Family support may be concentrated among those rich in other support sources Moderate levels of peer and significant-other support may play a protective role	High family support is significantly associated with less hopelessness, loneliness, depression, anxiety, somatization, suicidality, global severity and symptoms of mood and depressive disorder Supportive peer and other relationships associated with significantly less loneliness
Mohr and Sarno, 2016 USA The Ups and Downs of Being Lesbian, Gay, and Bisexual: A Daily Experience Perspective on Minority Stress and Support Processes.	Quantitative Daily diary methods recorded experiences for 7–10 days Measures included identity-salient experiences, proximal minority stress, affect	Participants (*n* = 61 students) Aged 17–28 years (mean 23.4) Self-identified as lesbian, gay and bisexual	Peer Positive and negative identity-salient experiences (ISEs) with heterosexual and LGB peers Ratios of positive to negative experiences were 1:1 for ISEs involving heterosexuals but 3:1 for ISEs involving other LGB peers	Increased positive affect on days featuring positive ISEs Decreased internalized stigma on days featuring positive ISEs with heterosexual peers Improved affect on days when levels of internalized stigma and expected rejection were lower than usual
Taliaferro et al., 2016 USA Nonsuicidal Self-Injury and Suicidality Among Sexual Minority Youth: Risk Factors and Protective Connectedness Factors.	Quantitative In-person survey data from 2013 Minnesota Student Survey of 9th and 11th grade students (*n* = 79,339) Measures included: connectedness: parent, teacher, friends, non-parent adults; school safety/risk; mental health: non-suicidal self-injury (NSSI) suicidality, depression, anxiety	Participants (*n* = 2223 youth) 14–18 years old (mean 15.5) Self-identified as lesbian, gay, bisexual, questioning (LGBQ) This sub-sample was more likely to identify as female, in grade 11 and white	Parent: Parent connectedness Provider: Other important connectedness: teacher, non-parent adults	Parental connectedness was significantly protective for all mental health outcomes Teacher caring, connectedness to other nonparental adults reduced risk of NSSI, suicidality and depressive symptoms School safety emerged as significant protective factor Effects were not as strong among bisexual youth
Watson et al., 2016 USA Sources of Social Support and Mental Health Among LGB Youth.	Quantitative Part of longitudinal study with data from wave 1) In-person survey Measures included social supports: close friends, teachers, classmates and parents; mental health: depression and self-esteem	Participants (*n* = 835 youth) 15–21 years old (mean 18.8) Self-identified as lesbian, gay, bisexual	Peer: LGB youth rated friend support as most prevalent and important Parent: Parent support rated lower but was very important	Parent support associated with higher self-esteem and lower depression for gay and bisexual male youth Parent, classmate and close friend support associated with less depression for lesbians, but not self-esteem Close friend support associated with less depression, and parent support associated with higher self-esteem for bisexual females
Wilson, 2016 USA The Impact of Discrimination on the Mental Health of Trans * Female Youth and the Protective Effect of Parental Support.	Quantitative Part of a wider study of trans * female youth) In-person survey Measures included psychological distress, depressive symptoms, post-traumatic stress (PTSD), stress related to suicidal thoughts; resiliency promoting factors, perceived social support, parental acceptance and closeness	Participants (*n* = 216 youth) 16–24 years old (mean 21.4) Self-identified as identified as female, transgender, genderqueer	Parent: Parental closeness Parental acceptance	Parental closeness was related to significantly lower odds of psychological distress, depressive symptoms, PTSD, stress related to suicidal thoughts Parental closeness resiliency promoting factor Higher reported resiliency associated with lower odds of psychological distress Higher parental acceptance of trans identity significantly lowered odds of PTSD
Kanhere et al., 2015 USA Psychosexual Development and Quality of Life Outcomes in Females with Congenital Adrenal Hyperplasia	Quantitative Survey Measures included family support, duration of care, self-satisfaction, body satisfaction, quality of life	Participants (*n* = 27 youth) 14–26 years old (66% ≤25) Self-identified as people with congenital adrenal hyperplasia (CAH)	Parent: Family social and psychological support Provider: Quality of care	Family and other social and psychological supports associated with positive perspectives during childhood and better quality of life during young adulthood
Watson et al., 2015 USA How Does Sexual Identity Disclosure Impact School Experiences?	Quantitative Survey as part of the Preventing School Harassment Survey (PSH) total sample (*n* = 1031) Measures included: academic achievement, being out to others, harassment at school	Participants (*n* = 375 youth) 12–18 years old (mean 15.7) Self-identified as lesbian, gay, bisexual	Parent/Peer: 18% were out to parents 41.5% were out to friends Approximately 19% were out to others at school More females disclosed their sexual identities to their parents More males were out to others at school	Youth who were not out at all or out to everyone had to manage their “outness” least Findings were more complex for youth who had to manage being out to different combinations of targets of disclosure Being out to more friends solely or in combination with others was generally associated with higher grades and less school harassment
Simons et al., 2013 USA Parental Support and Mental Health Among Transgender Adolescents	Quantitative Survey data from a larger study on the impact of a treatment protocol for transgender youth healthcare at Children’s Hospital LA Measures included parental support, depression, quality of life, perceived burden associated with being transgender, life satisfaction	Participants (*n* = 66 youth) 12–24 years (mean 19.1) Self-identification of internal gender identity different to that assigned at birth Wish to transition to desired sex	Parent: Receiving emotional help and support from parent Facilitating access to healthcare	Significantly associated with higher life satisfaction, lower perceived burden and fewer depressive symptoms Associated with higher quality of life, protective against depression Parents may have crucial role in offsetting mental health impact of societal harassment and discrimination
Mustanski et al., 2011 USA Mental Health of Lesbian, Gay, and Bisexual Youths: A Developmental Resiliency Perspective	Quantitative Survey of youth w someone accepting of their LGB orientation. Measures included: psychological distress, victimization, family support, peer support	Participants (*n* = 425 youth) 16–24 years old (mean 19.3) Self-identified as gay, lesbian and bisexual	Parent: Family support had significant promotive effects The positive effects of family support decreased with age Peer: Promotive effects of peer support increased with age	Peer support associated with lack of social loneliness, acceptance of sexual orientation and sense of having friends as a resource Peer support strongest correlate of psychological distress and promotive effect Social support did not ameliorate negative effects of victimization Increased resilience may suggest presence of resources
Bauermeister et al., 2010 USA Relationship Trajectories and Psychological Well-Being Among Sexual Minority Youth	Quantitative Survey using a structured interview protocol for Time 1 and Time 2 data Measures included psychological wellbeing (symptoms of depression, anxiety and internalized homophobia and self-esteem), relationships (dating relationships, sexual attraction), social support, disclosure of sexual identity	Participants (*n* = 350 youth) 15–19 years old (mean 17. 0) Self-identified as mostly gay or lesbian 75% categorized their level of same-sex attraction as “very” or “extremely” attracted	Peer: Involvement in a same-sex relationship Over a third of youth reported currently being in a same-sex relationship at both Time 1 and Time 2	Positively associated with changes in self-esteem in males Negatively correlated with changes in internalized homophobia in females
Doty et al., 2010 USA Sexuality Related Social Support Among Lesbian, Gay, and Bisexual Youth	Quantitative Survey-administered questionnaire battery Measures included sexuality stress, emotional distress, sexuality related and non-sexuality-related social support	Participants (*n* = 98 youth) 18–21 years old (mean 19.5) Self-identified as lesbian, gay, bisexual, unlabelled None self-identified as transgender	Parent: Close family provided non-sexuality support and least sexuality support Peer: Heterosexual friends provided more non-sexuality support than sexuality support Sexual minority friends provided support for coping with sexuality stress and support for coping with other problems	Sexuality support attenuated association between experiences of sexuality stress and emotional distress Longer time since initial disclosure associated with higher levels of reported support from heterosexual friends Non-sexuality-related social support did not buffer effects of sexuality stress on emotional distress
Ryan et al., 2010 USA Family Acceptance in Adolescence and the Health of LGBT Young Adults	Quantitative Survey informed by participatory research with LGBT youth and families Measures included family acceptance, self-esteem	Participants (*n* = 245 youth) 21–25 years old (mean 22.8) Self-identified as lesbian, gay, bisexual, alternative sexual identity	Parent: Family acceptance through positive experiences comments, behaviours and interactions	Positively associated with all three measures of positive adjustment and health: self-esteem, social support and general health
Sheets and Mohr, 2009 USA Perceived Social Support from Friends and Family and Psychosocial Functioning in Bisexual Young Adult College Students	Quantitative Online survey of bisexual students across 32 university campuses in US. Measures included: general social support, sexuality-specific support, depression, life satisfaction, internalized bi-negativity	Participants (*n* = 210 youth) 18–25 years old (mean 21.0) Self-identified as bisexual	Parent: General family support Sexuality-specific family support Peer: General friend support Sexuality-specific friend support	Negatively associated with depression Positively associated with life satisfaction Negatively associated with internalized bi-negativity
Detrie and Lease, 2008 USA The Relation of Social Support, Connectedness, and Collective Self-Esteem to the PsychologicalWell-Being of Lesbian, Gay, and Bisexual Youth	Quantitative Online survey Measures included: social support, family and friends, social connectedness, collective self-esteem, psychological wellbeing	Participants (*n* = 218 youth) 14–22 years (mean 18.0) Self-identified as lesbian, gay, bisexual (LGB) youth Most reported having some level of “outness”	Parent: Social support from family Particularly important for younger LGB youth Peers: Social support from friends Increasing importance for older youth Perceived more social support from friends than from family with age	Significantly predicted aspects of psychological wellbeing: self-acceptance, positive relations with others, autonomy, environmental mastery, purpose in life and personal growth Social connectedness was significantly correlated with collective self-esteem and was related to all aspects of psychological wellbeing
Darby-Mullins and Murdock, 2007 USA The Influence of Family Environment Factors on Self-Acceptance and Emotional Adjustment Among Gay, Lesbian, and Bisexual Adolescents	Quantitative In-person survey of sexual orientation identity and emotional adjustment, and family relationships/support Measures included: self-acceptance of sexual orientation identity, emotional adjustment, family relationships, parental support	Participants (*n* = 102 youth) 15–19 years old (mean 17.1) Self-identified as lesbian, gay, bisexual, queer, bi-curious, other	Parent: Positive general family environment Positive parental attitudes towards homosexuality Peer: Participants connected to and supported by their GLB peers and the GLB community	Family environment and parental attitudes towards homosexuality predict emotional adjustment but not in self-acceptance of sexual orientation Support from peers and community may impact the influence that the family environment has on self-acceptance
Floyd et al., 1999 USA Gay, Lesbian, and Bisexual Youths: Separation-Individuation, Parental Attitudes, Identity Consolidation, and Well-Being	Quantitative Survey interview, administered in person Measures included young adult–parent relationships, wellbeing (self-esteem, symptom distress), sexual orientation identity consolidation, parent attitudes regarding sexual orientation	Participants (*n* = 72 youth) 16–27 years (mean 20.9) Self-identified as lesbian, gay, bisexual	Parent: Relationships with both parents were important Relationships with mothers were generally closer and more supportive than fathers Accepting parental attitudes of sexual orientation Freedom from conflictual thoughts, independence and greater autonomy	Accepting parental attitudes/greater independence predicted positive wellbeing Parental attitudes predicted greater consolidation of sexual orientation identity Greater self-esteem associated with closer relatedness, freedom from conflictual thoughts, independence and autonomy Lower levels of symptom distress associated with more positive relatedness
Savin-Williams, 1989 USA Parental Influences on the Self-Esteem of Gay and Lesbian Youth: A Reflected Appraisals Model	Quantitative Survey of gay and lesbian youth, administered in person Measures included: parental importance, parental acceptance, comfortableness, self-esteem	Participants (*n* = 317 youth) Aged 14–23 years old 77% described themselves as predominantly or exclusively homosexual; 23% expressed some heterosexual interest also claiming to be gay or lesbian Two thirds gay, one third lesbian	Parent: Acceptance of young person’s sexual orientation Importance of the parental relationship for youth Intercorrelation between acceptance and importance	For lesbian youth, acceptance associated with comfort with sexual orientation, and importance increased positive associations between father acceptance and self-esteem For gay males, acceptance significantly predicted comfort, if parents were important, which was associated with positive self-esteem
Savin-Williams, 1989 USA Coming Out to Parents and Self-esteem Among Gay and Lesbian Youths	Quantitative Survey of gay and lesbian youth, administered in person Measures included: self-esteem, parental knowledge of identity, satisfaction with maternal/paternal relationship, contact with parents, marital status and age of parents	Participants (*n* = 317 youth) Aged 14–23 years old 77% described themselves as predominantly or exclusively homosexual; 23% expressed some heterosexual interest also claiming to be gay or lesbian Two thirds gay, one third lesbian	Parent: Knowing the sexual orientation of the child, frequent contact, satisfaction with the relationship and parents age were all highly correlated with each other Lesbians reported greater satisfaction with, and more contact with, mothers Gay youth were more out and had more parental contact, correlated with satisfaction	Lesbian youth reporting a satisfying relationship with mothers had the highest self-esteem/positive self-image Gay males out to mothers, and satisfying but infrequent relationship with fathers, more likely to report high self-esteem Differences in age, hometown community, occupational family status and sexual orientation

**Table 3 ijerph-18-11682-t003:** Qualitative, mixed-methods research and systematic review records of interpersonal relations: parental, peer and provider protective factors for LGBTI+ youth wellbeing: (*n* = 9).

Author/Year/Location Title	Methodology/Analysis	Demographic Details	Protective Factors	Wellbeing Indicator
Johnson et al., 2020 USA Trans Adolescents’ Perceptions and Experiences of Their Parents’ Supportive and Rejecting Behaviors.	Qualitative Interviews using lifeline methods/photo elicitation Recruited trans adolescents out to parents via another study of trans identity Analysis identified supporting, rejecting and mixed parental behaviours	Participants (*n* = 24 youth) 16–20 years old (mean 17.8) Self-identified as trans female, trans male, female, male, non-binary, genderqueer, genderfluid, non-binary trans guy, two-spirited, genderfluid transman, agender, non-binary trans masculine and gender nonconforming	Parent: Identity affirmation Self-education Emotional support Advocacy Instrumental support Assistance in obtaining gender affirming medical care	Increased positive wellbeing Potential for depression to lessen and hope for future selves to increase Improved ability to make important life decisions Enhanced active participation in their communities Facilitated development of internal resilience and ability to better endure stressors
McDermott et al., 2019 England Family trouble: Heteronormativity, Emotion Work and Queer Youth Mental Health.	Qualitative (two phase study) Phase 1 exploratory visual, creative and digital methods/interviews (youth *n* = 13, family member/mentor *n* = 7) Phase 2 diary methods/follow-up interviews (*n* = 9) Analysis included identification of how family relationships foster and maintain mental health and wellbeing of LGBTQ+ youth	Participants (*n* = 13 youth) 16–25 years old (mean 21.3) Self-identified as lesbian, gay, bisexual, pansexual and queer, other Participants defined their gender identity as trans female, trans male, (cis) female, (cis) male, other	Parent: Importance of family relationships for wellbeing Supported to explore identities in a safe environment while maintaining family bonds Emotionality of family relationships, and LGBTQ+ youth negotiation of these Time, respect and space to develop autonomy and self-determination	Belonging, security and becoming Disclosure of sexual and/or gender diversity crucial to good mental health Emotion work as a form of youth agency in relationship maintenance, endurance, repair and re-negotiation Maintained familial bonds through competency, self-awareness and compassion to family members
Bry et al. 2017 USA Management of a Concealable Stigmatized Identity: A Qualitative Study of Concealment, Disclosure, and Role Flexing Among Young, Resilient Sexual and Gender Minority Individuals	Qualitative Part of a larger longitudinal study (*n* = 450) Semi-structured interviews with resilient sexual and gender minority (SGM) youth Analysis identified social support networks, attitudes toward identities, discrimination, coping behaviours, coming out and family response	Participants (*n* = 10 youth) 18–22 years old (mean 20.2) Self-identified as gay male, bisexual male, gay, transgender female and bisexual, transgender female	Parent/peer: Disclosure of sexual orientation and gender identity Concealment of sexual orientation and gender identity Devaluing societal acceptance, perceived social support, trustworthiness Role flexing and individual identity management strategies	Coming out may increase open communication, structural social support and emotional support Desire for sense of authenticity, readiness, comfort with identity, personal safety Disclosure may reduce stigma and discrimination Concealment may increase unique strategies of accessing social support
Mehus et al., 2017 USA/Canada Living as an LGBTQ Adolescent and a Parent’s Child	Qualitative Part of a larger mixed-methods, multisite study Go-along interviews in which participants were accompanied Analysis identified factors that provide supportive LGBTQ youth environments	Participants (*n* = 66 youth) 14–19 years old (mean 16.6) Self-identified as gay/lesbian, bisexual, trans, queer or additional or other labels including: pansexual, rainbow sexual genderqueer, non-binary	Parent: Facilitated a loving, safe and accepting environment Supportive behaviours in day-to-day interactions: seeking to know how to be supportive Facilitated and connected youth to external resources through other supportive adults	Perceived parents’ love and acceptance promoted wellbeing, and increased likelihood of including parents in LGBTQ identities, sharing information/helping parents learn about LGBTQ issues Both reduced the need for external support and enhance access to external support Overlap in youth’s LGBTQ and family experience, which influenced interactions with social environment
Weinhardt et al., 2017 USA The Role of Family, Friend, and Significant Other Support in Well-Being Among Transgender and Non-Binary Youth.	Mixed-methods research In-person survey and one qualitative focus group Measures included living as one’s affirmed gender, social support, finding meaning in life, quality of life, mental health and resilience	Participants (*n* = 157) youth survey; focus groups (*n* = 8) Aged 13–21 years (mean 17.4) Self-identified as agender, transgender male/female, transgender, intersex, gender nonconforming, genderqueer, gender-fluid, non-binary, other gender category	Parent: Family support beyond acceptance to include advocacy, correct pronoun use and access to healthcare Peer: Friend support	Family support positively associated with living as one’s affirmed gender Reduced likelihood of experiencing a mental health issue in past year Positively associated with quality of life Friend support enhanced connectedness, pride and meaning in life
Mulcahy et al. 2016 USA Informal Mentoring for Lesbian, Gay, Bisexual, and Transgender Students.	Qualitative Semi-structured interviews Analysis identified experience of informal mentoring	Participants (*n* = 10) youth Aged 16–22 years (mean 18.0) Self-identified as lesbian, gay, bisexual, transgender	Provider: Informal mentor relationships with people who were good listeners, open-minded and non-judgemental Access to information and resources Facilitating social interactions	Improved self-awareness, confidence, comfort with sexual orientation Lessened isolation and loneliness at school Increased school safety and school engagement
Bouris et al., 2010 USA A Systematic Review of Parental Influences on the Health and Well-Being of Lesbian, Gay, and Bisexual Youth: Time for a New Public Health Research and Practice Agenda	Systematic review Search conducted across five databases Inclusion criteria: empirical peer-reviewed quantitative articles published between 1980 and 2010 Review investigated parental influences on lesbian, gay bisexual (LGB) youth health and wellbeing	Included studies (*n* = 31 records) Sample sizes ranged from *n* = 72 to *n* = 21,927 LGBT+ participants ranged from 10 to 24 years Parental influences on the mental health and wellbeing of LGB youth (*n* = 16), parental victimization of LGB youth (*n* = 1), parental influence on LGB youth’s experiences with suicide (*n* = 14)	Parent: Strong parent–child attachment characterised by closeness, support and connection Emotional dimensions of the parent–child relationship marked by knowledge of, and caring responses to, their child’s sexual orientation	Mediated the relationship between sexual orientation and mental health—depression, psychological distress Protective association with suicide Partially mediated the association between sexual orientation and suicidal tendencies
Diamond and Lucas, 2004 USA Sexual-Minority and Heterosexual Youths’ Peer Relationships: Experiences, Expectations, and Implications for Well-Being	Mixed-methods research Survey questionnaire and qualitative telephone interviews Total sample (*n* = 125) Measures included: outness, social networks (friendship experiences/expectations, romantic experiences, connectedness), mental health (depression, self-esteem and wellbeing) Analysis compared wellbeing of sexual minority youth (SMY) who were “out”, closeted or heterosexual	Participants (*n* = 60) 15–23 years old (mean 18.4) SMY self-identified as lesbian, gay, bisexual, unlabelled Females (*n* = 32) first knew they were not heterosexual at mean age 12.2, compared with the 10.3 mean years among the males (*n* = 28) 63% were out to their parents Nearly 70% were “out” to at least five heterosexual friends	Peer: Supportive peer relationships SMY “out” to more peers had larger networks, greater proportion of extremely close friends and more friendship loss/romantic relationship fears Younger SMY had smaller overall peer networks than young male heterosexuals, reported more friendship loss SMY reported disproportionately high worries about losing friends, low feelings of control over romantic relationships	Supportive peer relationships may be particularly important for SMY and directly related to psychological wellbeing SMY had comparable self-esteem, mastery and perceived stress as did heterosexuals, but greater negative affect SMY had similar peer connectedness, perceived control and similar romantic relationships to heterosexual peers Outness was found to be “neither uniformly positive nor uniformly negative”
Galupo and St John, 2001 USA Benefits of Cross-Sexual Orientation Friendships Among Adolescent Females	Qualitative Semi-structured interviews conducted close cross-sexual orientation friendship pairs Joint interviews (*n* = 10) and individual interviews with each (*n* = 20) Analysis identified the mutual benefits for lesbian, bisexual and heterosexual youth	Participants (*n* = 20 youth 18–25 years old (mean 19.5) Self-identified as lesbian (*n* = 5) bisexual (*n* = 5), heterosexual (*n* = 10) Only two had disclosed their sexual orientation to their parents	Peer: Mutually beneficial cross-sexual orientation friendships Appreciation for commonalities Objectivity in life Increased sensitivity to sexual minority perspectives Breaking down negative stereotypes	Increased self-acceptance and self-esteem for lesbian and bisexual youth For heterosexual young women, increased flexibility in the understanding of personal sexual identity Increased closeness and trust within the friendship

**Table 4 ijerph-18-11682-t004:** Quantitative, mixed-methods research and systematic review of records of community protective factors for LGBTI+ youth wellbeing: LGBTI+, online, faith and cultural communities (*n* = 12).

Author/Year/Location Title	Methodology/Analysis	Demographic Details	Community Protective Factors	Wellbeing Indicator
Eisenberg et al., 2020 USA LGBTQ Youth-Serving Organizations: What Do They Offer and Do They Protect Against Emotional Distress?	Quantitative Online/pencil + paper survey Data merged from 2013 Minnesota Student Survey and the LGBTQ Environment Inventory Measures included internalised symptoms, self-harm and suicidal ideation or attempt	Participants (*n* = 2454 youth) 13–16 years old (mean 14.3) Self-identified as lesbian, gay, bisexual or unsure (questioning)	LGBTI+ communities The presence of LGBQ organizational and community resources (rather than direct involvement in activities or programs) is protective	Living in areas with LGBTQ organizations and community resources associated with lower odds of emotional distress The protective factor was greater for girls
McCann et al., 2020 Global An Exploration of the Relationship Between Spirituality, Religion and Mental Health Among Youth Who Identify as LGBT+: A Systematic Literature Review	Systematic review Search conducted across four databases Inclusion criteria: empirical peer-reviewed research in English on mental health and spirituality or religious experiences of LGBT+ youth	Included studies (*n* = 9 records) Sample sizes ranged from *n* = 1 to *n* = 1413 LGBT+ participants ranged from 12 to 25 years Quantitative (*n* = 5), qualitative (*n* = 2), mixed-methods (*n* = 2)	Faith communities Presence of accepting faith community Potential for faith communities to be a source of support	Potential for acceptance and support Connection with a higher power Some youth find other ways of reconciling and constructing spiritual and LGBT+ identities
Wagaman et al., 2020 USA Managing Stressors Online and Offline: LGBTQ+ Youth in the Southern United States	Quantitative Online survey data from larger MMR study Measures included: LGBTQ esteem, offline and online support	Participants (*n* = 662 youth) 14–29 years old (mean 18.1) Self-identified as: lesbian, gay, bisexual, trans, queer, questioning, asexual, gender queer/gender fluid, non-binary, two-spirit, pansexual	Online communities Online platforms facilitate access to LGBTQ+-specific social support	Significantly moderated the impact of LGBTQ-specific stressors on esteem May be protective for youth not connected to school/community-based resources and services
McInroy, 2019 USA/Canada Building Connections and Slaying Basilisks: Fostering Support, Resilience, and Positive Adjustment for Sexual and Gender Minority Youth in Online Fandom Communities	Mixed-methods research Online survey data and qualitative data from long answer questions re. online community and sexual and gender (SGM) identity Measures included: SGM identity and development, technology use, health and mental health, community engagement	Participants (*n* = 3665 youth) 14–29 years old (mean 17.8) Self-identified as pansexual/panromantic, bisexual/biromantic, queer, asexual/aromantic, non-binary/independent, lesbian, gay, queer/genderqueer, trans * man/male and trans * woman/female	Online communities May increase connectedness Source of social support Provide opportunities for mentorship	Facilitate navigation of challenges Foster resilience Encourage positive adjustment Affirm feelings of strength
Rubino et al., 2018 Australia Internalized Homophobia and Depression in Lesbian Women: The Protective Role of Pride	Quantitative Survey data from lesbian women across the states of Victoria and NSW Measures included: self-disclosure, internalised homophobia, self-esteem, collective self-esteem and depression	Participants (*n* = 225 adults) 18–62 years old (mean 23.2) Self-identified as lesbian	LGBTI+ communities (lesbian) Combined collective self-esteem, self-esteem and self-disclosure represented a broader concept of pride Pride reflects individual self-esteem, group membership, and “outness”	Pride is significantly associated with an inverse relationship between self-esteem and depression in lesbian women Pride mediates the relationship between internalized homophobia and depression
Scroggs et al., 2018 USA Identity Development and Integration of Religious Identities in Gender and Sexual Minority Emerging Adults	Quantitative Survey data from the Social Justice Sexuality Project. Oversampling methods were utilized in order to increase gender and sexual minority (GSM) people of colour Measures included: identity salience, integration and visibility, GSM activity, religious activity, wellbeing	Participants (*n* = 961 youth) 18–24 years old (mean 21.0) Self-identified as lesbian, gay, bisexual, transgender, another gender identity, queer, same-gender loving	LGBTI+ communities GSM group activities associated with identity visibility (outness) Identity visibility positively associated with increase in GSM activity Faith communities Religious group activity is associated with identity development and integration	Increases in GSM group activity are associated with wellbeing Religious group activity is associated with increases in wellbeing Religious group activity mediates the relationship between identity integration and wellbeing
Ceglarek and Ward, 2016 USA A Tool for Help or Harm? How Associations Between Social Networking Use, Social Support, and Mental Health Differ for Sexual Minority and Heterosexual Youth	Quantitative Online survey Measures included: social support, lesbian, gay and bisexual identity development, social networking site use, mental health and wellbeing	Participants (*n* = 146 youth) 18–24 years old (mean 20.2) Self-identified as: lesbian, gay and bisexual	Online communities Sexual minority youth use sites specifically for sexual identity development	Enhanced social communication Predicted positive mental health outcome
Meanley et al., 2016 USA Psychological Well-being Among Religious and Spiritual-identified Young Gay and Bisexual Men	Quantitative Online survey examining structural/psychosocial vulnerabilities experienced by male sexual minority youth (SMY) in Detroit Measures included: religious commitment, participation and coping, self-esteem, life purpose, internalised homophobia, stigma	Participants (*n* = 397 people) 18–29 years old (mean 23.2) Self-identified as gay/homosexual, bisexual, same gender loving, MSM or other	Faith communities 80% of the sample identified as religious/spiritual Most (91%) identified spirituality as a coping source Challenges remain in reconciling potentially conflicting identities	Spiritual coping had a protective association on life purpose and self-esteem Connection with one’s spirituality may be a source of strength Spirituality may foster resilience
Zimmerman et al., 2015 USA Resilience in Community: A Social Ecological Development Model for Young Adult Sexual Minority Women	Quantitative Baseline and 12-month online survey with sexual minority women (SMW) about family support and rejection Measures: age of coming out, LGB Identity Scale, Outness Inventory, family rejection, Connectedness to the LGBTQ Community Scale, Collective Self-Esteem Scale	Participants (*n* = 843 youth) 18–25 years old (mean 21.4) Self-identified as lesbian and bisexual, with 57% identified as bisexual	LGBTI+ communities (lesbian) Connection to sexual minority communities was greater for those experiencing family rejection Increased stigma and concealment motivation did not impact community connectedness	Increased self-esteem Enhanced resilience Racial minority SMW reported collective self-esteem
Gattis et al., 2014 USA Discrimination and Depressive Symptoms Among Sexual Minority Youth: Is Gay-Affirming Religious Affiliation a Protective Factor?	Quantitative Cross-sectional survey data on campus climate and religious affiliation (total sample *n* = 2120) Measures included: depressive symptoms, religious affiliation, denomination affirmation of same-sex marriage	Participants (*n* = 393 people) 18–28 years (mean 23.4) Self-identified as “completely gay/lesbian”, “mostly gay/lesbian”, bisexual and mostly heterosexual	Faith communities Religious affiliation with a gay-affirming denomination, i.e., endorsing same-sex marriage)	Reduced the harmful effects of discrimination amongsexual minority youth
Longo et al., 2013 USA Religion and Religiosity: Protective or Harmful Factors for Sexual Minority Youth?	Quantitative Online survey with gay, lesbian, bisexual, transgender, questioning or queer (LGBTQ) youth in Colorado Measures included: psychosocial risk factors for self-harming behaviour and religious tradition/religiosity	Participants (*n* = 250 youth) 13–25 years old (mean 16.8) Self-identified as gay, lesbian, bisexual, pansexual, queer, asexual, other and not sure/questioning	Faith communities Religion potentially plays both a protective and harmful role for LGBTQ youth General coping value of religion for some youth Support offered through a religious framework may be beneficial for some youth	Those who reported being Christian with little to no or some religious guidance had the least risk of self-harming behaviours Religion may be meeting existential needs for Christians with low religious guidance
Walker and Longmire-Avital, 2013 USA The Impact of Religious Faith and Internalized Homonegativity onResiliency for Black Lesbian, Gay, and Bisexual Emerging Adults	Quantitative Online survey of Black LGB emerging adults on religious faith and psychological wellbeing Measures included: religious faith, resiliency, internalized homonegativity, mental health (anxiety/depression)	Participants (*n* = 175 youth) 18–25 years old (mean 21.3) Self-identified as lesbian, gay and bisexual	Faith communities For Black LGB emerging adults, sexual minority identity and religiosity are not mutually exclusive	Religious faith was significant contributor to resiliency Religious faith played significant role in coping with adversity Participants with a college degree or more were significantly more resilient and less depressed

**Table 5 ijerph-18-11682-t005:** Qualitative records of community protective factors for LGBTI+ youth wellbeing: LGBTI+, online, faith and cultural communities (*n* = 12).

Author/Year/Location Title	Methodology/Analysis	Demographic Details	Community Protective Factors	Wellbeing Indicator
Goffnett et al., 2020 USA Challenges, Pride, and Connection: A Qualitative Exploration of Advice Transgender Youth Have for Other Transgender Youth	Qualitative Face-to-face and online interviews Analysis identified three themes promoting trans youth wellbeing: challenges are real; pride; you are not alone	Participants (*n* = 19 youth) 15–22 years old (mean 18.2) Self-identified as transgender man/masculine, non-binary/gender fluid, transgender woman/feminine	LGBTI+ communities (trans) Opportunities for social connections Accepting support network	Validation of identity Finding positives Maintaining perspective of challenges as temporary Perseverance despite challenges Cultivating hope for the future
Paceley et al., 2020 USA “Sometimes you get married on Facebook”: The Use of Social Media among Nonmetropolitan Sexual and Gender Minority Youth	Qualitative In-depth interviews (part of larger MMR study) Grounded theory analysis identified three categories of online use	Participants (*n* = 34 youth) 14–18 years old (mean 16.0) Self-identified as SGM: bisexual, pansexual, gay, lesbian and queer, transgender, questioning	Online communities Important platform for nonmetropolitan SGM youth Particularly protective if no supports in local communities Access SGM identified people, resources and information	Establish a sense of community Friendships and relationships Establish SGM support Space/platform for self-expression when coming out Space/platform for venting
Selkie et al., 2020 USA Transgender Adolescents’ Uses of Social Media for Social Support	Qualitative Semi-structured interviews Analysis identified four types of support for trans adolescents	Participants (*n* = 25 youth) 15–18 years old (mean 16.0) Self-identified as transfeminine, transmasculine and non-binary	Online communities Emotional support through peers and role models Appraisal support for validating experiences Informational support for navigating health decisions and educating family and friends	Recognition of selves in transgender peers Receive affirmation and validation through positive feedback Promote support-seeking behaviours online Increased self-esteem Improved navigation and acceptance of identity
Chiang et al., 2019 New Zealand Navigating Double Marginalisation: Migrant Chinese Sexual and Gender Minority Young People’s Views on Mental Health Challenges and Supports	Qualitative Face-to-face semi-structured interviews with Chinese sexual/gender minority (SGM) people residing in Auckland, New Zealand Analysis identified intersecting identities, including supporting and resiliency factors	Participants (*n* = 11 youth) 19–29 years old (mean 23.3) Self-identified as lesbian, gay, bisexual, transgender, asexual, questioning, undecided, no label	Cultural communities Helpful Chinese cultural factors, including strong cultural and familial ties Unconditional love of parents Support from peers and inspiration of role models	Strong cultural ties and family ties enhanced personalised coping strategies Good work ethic Access to professional therapeutic support
Schmitz et al., 2019 USA LGBTQ+ Latinx Young Adults’ Health Autonomy in Resisting Cultural Stigma	Qualitative In-depth face-to-face interviews Sexual and gender minority (SGM) Latinx youth in the border region of Texas Analysis identified three protective factors promoting and fostering health	Participants (*n* = 41 youth) 18–26 years old (mean 21.0) Self-identified as lesbian, gay, bisexual, transgender, queer, other, non-binary	Cultural communities Personal networks: family, friends and health and social care providers Trusted friends and family as a source of support and information Information seeking through online social networks	Positive LGBT identity and cultural identity Health autonomy Resistance to cultural stigma and prejudice related to intersecting identities Challenged “at risk framing” of cultural messages
Morris, 2018 UK “Gay capital” in GayStudent FriendshipNetworks: AnIntersectional Analysis ofClass, Masculinity, andDecreased Homophobia	Qualitative In-depth, face-to-face semi-structured interviews with gay male youth from four universities across England Analysis explored the dynamics of friendship networks in the context of decreased homophobia	Participants (*n* = 40 youth) 18–21 years old (mean 19.5) Self-identified as gay	LGBTI+ communities (gay) Environment with decreased homophobia In-person social networks Gay and straight peer friendships	Gay capital Shared knowledge of gay cultures Belonging to gay social networks Recognition of gay identity as a form of prestige
Wolowic et al., 2018 USA/Canada Come Along With Me: Linking LGBTQ Youth to Supportive Resources	Qualitative Part of a larger mixed-methods, multisite study in Minnesota, Massachusetts and British Columbia. Go-along interviews in which participants were accompanied Analysis identified factors that provide supportive LGBTQ youth environments	Participants (*n* = 66 youth) 14–19 years old (mean 16.6) Self-identified as gay/lesbian, bisexual, trans, queer or additional or other labels including pansexual, rainbow sexual genderqueer, non-binary	LGBTI+ communities Indirect links, such as LGBTQ media and print advertising Personal links, including referrals to LGBTQ organizations from trusted friends or adults Regular attendance at LGBTQ programs	Affirmation of LGBTQ identities Increased awareness of supports and resources May assist in forming denser networks of support Prompted self-agency and integration into supportive environments
Zeeman et al., 2017 UK Promoting Resilience and Emotional Well-Being of Transgender Young People: Research at the Intersections of Gender and Sexuality	Qualitative Focus group discussion (*n* = 19), including a focus group with trans youth (*n* = 1) Analysis identified individual and collective capacities and resources that support resilience and wellbeing	Participants (*n* = 5 youth) 14–19 years old (mean 16.2) Self-identified as transgender	LGBTI+ communities (trans) Safe spaces for connection and shared activities Trans youth club attendance	Facilitates mutual trust and support Feeling safe and connected to others, despite adversity Increased self-confidence
Rios and Eaton, 2016 USA Perceived Social Support in the Lives of Gay, Bisexual and Queer Hispanic College Men	Qualitative Face-to-face, semi-structured interviews with sexual minority men (SMM) college students in New England and Southeast USA about support Analysis identified four types of support	Participants (*n* = 51 students) 18–35 years (mean 21.5) Self-identified as gay or homosexual, bisexual or other: heteroflexible or “open”	LGBTI+ communities (gay) Connected with lesbian, gay and bisexual communities through shared experiences Connected through strongholds of support to LGB communities Those in leadership positions cultivated climates of support	Support increased gradually and over time Experience of endorsement and caring Psychologically, emotionally or physically protected Enabled successful navigation of identity development
Craig et al., 2015 Canada Media: A Catalyst for Resilience in Lesbian, Gay, Bisexual, Transgender, and Queer Youth	Qualitative Face-to-face, in-depth interviews using grounded theory with LGBTQ youth on positive media representation Analysis identified four forms of protection	Participants (*n* = 19 youth) 18–22 years old (mean 19.5) Self-identified as lesbian, gay, bisexual, transgender, queer, gender queer, questioning, pansexual	Online communities Use of online social media and visibility in the media were protective Fostering a sense of community	Connection with LGBTQ communities Means of escaping their discriminatory reality Regaining strength after negative experiences Facilitating advocacy and resistance
Singh, 2013 USA Transgender Youth of Color and Resilience: NegotiatingOppression and Finding Support	Qualitative Face-to-face interviews of transgender youth of colour, from a large south-eastern US city, self-described as resilient Thematic analysis identified protective factors across five domains	Participants (*n* = 13 youth) 15–24 years old (mean 19.0) Self-identified trans-masculine and trans-feminine, and 3 did not relate to the use of these terms to describe their gender expressionsor identities (genderqueer or gender-fluid)	LGBTI+ communities (trans) Connecting to and expressing cultural/gender identities Access to supportive systems/providers: education, health Reframing mental health challenges Visibility of other youth	Affirmation of one’s identities through the value of unique and multiple identities Self-definition helped to instil a sense of pride and increased self-acceptance Enhanced ability to more closely connect Resilience, liberation and empowerment
Harper et al., 2012 USA What’s Good About Being Gay? Perspectives from Youth	Qualitative Semi-structured, in-depth interviews with ethnically diverse gay and bisexual youth in Chicago and Miami Thematic analysis identified positive perceptions of gay and bisexual identity	Participants (*n* = 63 youth) 14–22 years old (mean 20.3) Self-identified as: gay and bisexual	LGBTI+ communities (gay) Connectedness to gay communities Connectedness to women Flexibility re. sexual orientation and gender norms and environmental flexibility through access to spaces	Positive personal conceptualizations of being gay/bisexual Resilience through acceptance, self-care, rejection of stereotypes and activism
Hillier et al., 2012 USA The Internet As a Safety Net: Findings From a Series of Online Focus Groups With LGB and Non-LGB Young People in the United States	Qualitative Online focus groups (*n* = 3), two with lesbian, gay and bisexual (LGB) youth and one non-LGB group Analysis identified protective factors from internet use for LGB youth	Participants (*n* = 33 youth) 13–18 years old (mean 15.5) Self-identified as lesbian, gay, bisexual, pansexual, queer	Online communities Access to online friendships, support from friends online, meeting people offline from the internet, finding romance online gay community online LGB youth were more adventurous in online use For some, online friendships replaced in-person friendships	Safe spaces to explore feelings and sexuality Accepting and supportive friendships Access to information about same-sex romance and relationships Finding discourses beyond those that pathologized LGB youth
Singh et al., 2012 USA “I Am My Own Gender”: Resilience Strategies of Trans Youth	Qualitative Face-to-face, semi-structured interviews with trans youth on resilience strategies for navigating stressors Analysis identified five forms of resilience strategies for navigating stressors	Participants (*n* = 19 youth) 15–25 years old (mean 22.0) Self-identified as trans man or trans guy, female to male, male, genderqueer, male to female Participants defined their sexual orientation as queer, gay, pansexual, asexual, straight, unreported	LGBTI+ communities (trans) Community connectedness Enhanced access to supports including educational, counselling and healthcare providers Navigation of relationships with family and friends	Affirmation of trans identity and individual journey Increased self-advocacy Proactive agency to access supportive systems and providers Reframing of mental health challenges Enhanced strategies of resilience
DiFulvio, 2011 USA Sexual Minority Youth, Social Connection and Resilience: From Personal Struggle toCollective Identity	Qualitative Face-to-face unstructured interviews (*n* = 22) and 2 focus group discussions (2) using life story methods with sexual minority youth (SMY) Analysis identified LGBQ participants experience of strengths and challenges	Participants (*n* = 22 youth) 14–22 years old (mean 18.0) Self-identified as lesbian, gay, bisexual, queer	LGBTI+ communities Social connection provided a forum for moving personal struggle to collective action Individual connection provided social networks Group affiliation provided affirmation of identity	Social connection contributes to resilience, self-acceptance, pride and a sense of regaining power over one’s life Acknowledgement of collective experience may assist in making meaning of LGBQ identity Connection gave a sense of purpose in addressing structural inequalities
Munoz-Plaza et al., 2002 USA Lesbian, Gay, Bisexual and Transgender Students: Perceived Social Support in the High School Environment	Qualitative Face-to-face interviews with undergraduate students attending universities in North Carolina Analysis identified available support systems for LGBT youth in the high school environment	Participants (*n* = 12 youth) 18–21 years (mean 19.0) Self-identified as: lesbian, gay, bisexual and undecided. No trans students participated	LGBTI+ communities LGBT-identified friends, peers and adults provided emotional, instrumental, informational and appraisal support “Close” friends were relied on most for emotional support around personal issues	Visibility enhanced comfort with own identity and disclosure to others Increased comfort and acceptance Limitations to the emotional support from heterosexual peers to whom they disclosed their orientation
Nesmith et al., 1999 USA Gay, Lesbian, and Bisexual Youth and Young Adults	Qualitative Interviews using open-ended questions with gay, lesbian or bisexual (LGB) youth in Seattle about perceived support Analysis identified four domains of protection for LGB youth	Participants (*n* = 17 youth) 15–22 years (mean 18.8) Self-identified as lesbian, gay and bisexual. None identified as transgender	LGBTI+ communities Access to concrete support Access to emotional support, specifically relating to sexual orientation Access to financial support Informational support around LGB issues Role models	LGB peers and adults were perceived as more supportive, particularly regarding informational support Acquiring a sense of community Locating parental figures among LGB adults

**Table 6 ijerph-18-11682-t006:** Records of legal protective factors for LGBTI+ youth wellbeing: inclusive policies, curriculum, access and provision (*n* = 18).

Author/Year/Location Title	Methodology/Analysis	Demographic Details	Protective Factors	Wellbeing Indicator
Poteat et al., 2019 USA Greater Engagement in Gender-Sexuality Alliances (GSAs) and GSA Characteristics Predict Youth Empowerment and Reduced Mental Health Concerns.	Quantitative Two-wave survey of students from 38 Gay–Straight Alliances (GSAs) Measures included GSA engagement level, perceived peer validation, self-efficacy to promote social justice, hope, depressive/anxiety symptoms	Participants (*n* = 580 youth) 10–20 years old (mean 15.6) Self-identified as lesbian, gay, bisexual, heterosexual, queer, questioning asexual, other or no response Participants defined their gender identity as cisgender female or male, non-binary, transgender, genderqueer, gender fluid, other or not reported	Gender Sexuality Alliances Greater engagement across school year in GSAs Enhanced access to support, information, resources and advocacy GSAs may meet the diverse needs across sexual orientation, gender identity or race/ethnicity	Through increased hope, greater engagement indirectly predicted reduced depressive and anxiety symptoms GSAs whose members had more mental health discussions and more meetings reported reduced mental health concerns
Weinhardt et al., 2019, USA Transgender and Gender Nonconforming Youths’ Public Facilities Use and Psychological Well-Being: A Mixed-Method Study	Mixed-methods research Gender Identity and Health Youth Survey (*n* = 127) and two focus groups (*n* = 9) Measures included self-esteem, resilience, quality of life, perceived stigma, feelings of safety, public facility use Analysis identified perceptions, attitudes, public bathroom access	Participants (*n* = 127 youth) 13 -20 years (mean 17.2) Self-identified as agender, transgender, gender nonconforming, genderqueer, non-binary, other and multiple gender identities	Inclusive policies Access to multiple-user bathrooms corresponding to gender identity must be accompanied by policies and actions that support those who use them Access to single-user bathrooms normalizes their use for all students, rather than singling out TGNC youth	Bathroom and locker room policy and practice re. access were associated with comfort, belonging and safety in school Feeling safe in bathrooms due to appearance or gender was associated with significantly higher levels of resilience Promoted youth agency
McDonald, 2018 Global Social Support and Mental Health in LGBTQ Adolescents: A Review of the Literature.	Narrative review Search conducted across three databases Inclusion criteria quantitative, journal articles in English, published 1982–2016, on effects of social support mental health for lesbian, gay, bisexual, transgender (LGBTQ) youth Analysis identified social support, networks and social connectedness as protective	Included studies Sample sizes ranged from *n* = 98 to *n* = 680 LGBT+ participants ranged from 13 to 23 years of age Multiple definitions of support including social support, support networks and connections to support groups, alongside family support	Gay-Straight Alliances: Presence and involvement Parent: Support from significant family members Provider: Support from respected adults	GSAs associated with higher levels of self-esteem, which in turn associated with wellbeing Family support associated with reduced mental health disorders, such as symptoms of depression, anxiety, suicidal ideation and suicide School support facilitated smoother school experience with less school avoidance/higher self-motivation
Russell et al., 2018 USA Chosen Name Use Is Linked to Reduced Depressive Symptoms, Suicidal Ideation, and Suicidal Behavior Among Transgender Youth.	Quantitative Survey data from the Risk and Protective Factors for Suicide Among Sexual Minority Youth Study Measures included depressive symptoms, suicidal ideation and behaviour, social support and chosen name use	Participants (*n* = 129 youth) 15–21 years old (mean 19.5) Self-identified as transgender and gender nonconforming youth	Inclusive policies: Chosen name use and pronouns in multiple contexts Policies that promote the social transition process of gender affirmation Parent/peer/provider: Chosen name use and pronouns	Affirmed gender identity An increase by one context in which a chosen name predicted a 5.37-unit decrease in depressive symptoms, a 29% decrease in suicidal ideation and a 56% decrease in suicidal behaviour
Porta, Gower et al., 2017 Canada/USA “Kicked out”: LGBTQ youths’ Bathroom Experiences and Preferences.	Qualitative Part of a larger mixed-methods, multisite study Go-along interviews in which participants were accompanied (*n* = 66) Analysis of responses by 25 youth (38%) who mentioned bathrooms during their interviews and were included in the analysis	Participants (*n* = 25 youth) 14–19 years old (mean 16.0) Self-identified as gay/lesbian, bisexual, queer or additional or other labels including pansexual, rainbow sexual genderqueer, non-binary Larger proportion in sub population self-identified as trans/fluid/gender neutral l/“other” than the overall sample (52% versus 32%)	Inclusive policies Provision and access to gender-neutral bathrooms Gay–Straight Alliances Presence and provision of GSAs Facilitated access to supportive adults	Fostered a sense of safety and inclusivity Sense of a welcoming environment A sense of and identity advocacy from adults Improved feelings of safety for LGBTQ youth has the potential to improve health equity
Porta, Singer et al., 2017 Canada/US LGBTQ Youth’s Views on Gay-Straight Alliances: Building Community, Providing Gateways, and Representing Safety and Support.	Qualitative Part of a larger mixed-methods, multisite study Go-along interviews in which participants were accompanied (*n* = 66) Analysis of responses of LGBTQ youth (88%) who mentioned Gay–Straight Alliances during interviews	Participants (*n* = 58 youth) 14–19 years old (mean 16.6) Self-identified as gay/lesbian, bisexual, trans, queer or additional or other labels including pansexual, rainbow sexual genderqueer, non-binary	Gay–Straight Alliances Supportive environments through provision of, and participation in, a GSA Opportunities for leadership Access to resources: supportive adults and informal social locations	Increased emotional connection Feeling of support/belonging Enhanced sense of community membership Increased sense of safety Improved outcomes for all students, not just those who identify as LGBTQ
Wernick et al., 2017 USA Gender Identity Disparities in Bathroom Safety and Wellbeing among High School Students	Quantitative Part of a multi-school climate survey Survey data on high school students’ wellbeing Measures included bathroom safety and wellbeing outcomes: school safety, self-esteem, grades	Participants (*n* = 86) 14–18 years old (mean 16.3) Self-identified as genderqueer, agender, questioning, transgender, gender nonconforming or a gender identity not listed Half (51%) defined their sexual orientation as LGBQ	Inclusive policies (bathrooms) Policies and practices that ensure students’ right to safely access bathrooms Recognition of bathroom access as one issue in a range of concerns LGBQ students may also benefit	Explicit support for trans students and feelings of safety in the bathroom associated with improved wellbeing Mediated the effect of trans identity on overall school safety Indirect effect on self-esteem
Jones, 2016 Australia Education Policies: Potential Impacts and Implications in Australia and Beyond.	Mixed-methods research Online survey from a 2010 national survey, interviews with policy informants (*n* = 10), documentary analysis (*n* = 80) Measures included: available supports and protections and outcomes such as self-harm, suicidality, school safety and support, feelings about sexuality	Participants (*n* = 3134 youth) 14–21 years old (mean 17.5) Self-identified as gay, lesbian, bisexual, transgender, intersex and queer	Inclusive policies Protective state-level protections mentioning GLBTIQ students explicitly LGBT school supports Inclusive curriculum Affirming and comprehensive sexuality education Inclusive extracurricular activities Equal treatment of same-sex partners at events	Increased feelings of safety and being protected at school Inclusive, affirming messages associated with feeling good about sexuality and gender identity Reduced likelihood of thinking about self-harm and decreased risks of suicidality and self-harm
Poteat et al., 2016 USA Promoting Youth Agency Through Dimensions of Gay-Straight Alliance Involvement and Conditions that Maximize Associations.	Quantitative Survey data from the 2014 Massachusetts Gay–Straight Alliance Network survey. Total sample (*n* = 295) Measures included: family support, GSA organizational structure, support/socializing, information/resources, advocacy, perceived positive school LGBT climate	Participants (*n* = 205 youth) 13 to 20 years old (mean 16.1) Self-identified as lesbian, gay, bisexual, questioning, other write-in responses including pansexual and queer Participants defined their gender as male, female, transgender (MtF), transgender (FtM), genderqueer, other responses	Gay–Straight Alliances Provision of support Opportunities to socialise Provision of information Access to resources	More organisational structure in GSA associated with increased advocacy Increased support/socializing and access to information/resources resulted in more advocacy Increased advocacy in GSAs associated with greater agency among sexual minority youth
Poteat et al., 2015 USA Contextualizing Gay-Straight Alliances: Student, Advisor, and Structural Factors Related to Positive Youth Development Among Members.	Quantitative In-person survey Total sample *n* = 146 Observation of Gay–Straight Alliance meetings (*n* = 13) Measures included: victimisation, GSA support and advocacy, mastery, sense of purpose and self-esteem	Participants (*n* = 85 youth) 14–19 years old (mean: 16.0) Self-identified as lesbian, gay, bisexual, questioning or other or not reported Participants defined their gender identity as female male, transgender (FtM), other or not reported	Gay–Straight Alliances Flexible GSA structures that balance support and advocacy Positive emotional climate Opportunities for leadership Diverse expressions of leadership: emotional support Perception of supportive school contexts	Higher perceptions of support associated with wellbeing Less advisor control predicted greater mastery GSA advocacy predicted sense of purpose Racial/ethnic minority youth reported greater wellbeing, yet lower support
Hatzenbuehler et al., 2014 USA Protective School Climates and Reduced Risk for Suicide Ideation in Sexual Minority Youths.	Quantitative Survey data Pooled from 2005 and 2007 Youth Risk Behavior Surveillance Surveys/2010 School Health Profile Survey Measures included school climate and suicidal thoughts, plans and attempts	Participants (*n* = 4314 youth) 14–18 years old (mean 15.5) Self-identified as lesbian, gay, bisexual, unsure	Gay–Straight Alliances Presence of GSAs is affirming Encourages school personnel to attend training Inclusive curriculum Sexuality puberty education Inclusive policies Policies that prohibit discrimination	GSAs provide safe spaces for LGBTQ youth and allies Protective school climates associated with fewer past-year suicidal thoughts Disparities in suicidal thoughts nearly eliminated in states with the most protective school climates
Heck et al., 2013 USA Offsetting Risks: High School Gay Straight Alliances and Lesbian, Gay, Bisexual, and Transgender (LGBT) Youth.	Quantitative Online survey Recruitment via college and university lesbian, gay, bisexual and transgender (LGBT) student organizations and a social networking site Participants were young adults, under 21 (the age of legal alcohol use), with 12 or more years of education Measures included school belonging, depression and general psychological distress	Participants (*n* = 145 youth) 18–20 years old (mean 19.2) Self-identified as lesbian, gay, other Participants defined their gender identity as transgender/other, female, male	Gay–Straight Alliances Presence of a GSA may be indicative of an environment that is conducive to healthy development for LGBT youth	Attendance at a high school with a GSA associated with significantly more favourable outcomes in relation to school experiences, alcohol use and psychological distress GSAs associated with significantly higher ratings of school belonging Community climate a significant predictor of school belonging
McCarty-Caplan, 2013 USA Schools, Sex Education, and Support for Sexual Minorities: Exploring Historic Marginalization and Future Potential	Narrative review Search conducted via Ebscohost search engine Inclusion criteria: literature, in English, published between 1987 and 2013 Analysis identified protective factors with a focus on schools improving capacity to support sexual minority youth	Studies Participants ranged from 18 to 24 years Sample sizes across articles not provided Review included journal articles, book chapters and reports	Gay–Straight Alliances: Establishing student groups as a means of structural support Inclusive policies: Mission statements and non-discrimination policies Positive representations Inclusive curriculum: Inclusive sexuality and relationship education Provider/peer: Role models in teaching or administrative roles Allies among student population	Inclusion associated with better adjustment to academic environments Promoted the value of inclusion and diversity Conveyed a strong public message of support and acceptance Identification with role models Enhanced sense of peer allyship and advocacy Improved school experiences and belonging
Jones and Hillier, 2012 Australia Sexuality Education School Policy for Australian GLBTIQ Students	Mixed-methods research Quantitative data from 2010 online national survey on Australian GLBTIQ young people Qualitative interviews (*n* = 8 policy informants) Documentary analysis (*n* = 80 texts) Measures included policy impacts on sexual and gender minority youth including sexuality education	Participants (*n* = 3134 youth) 14–21 years old (mean 17.5) Self-identified as gay/lesbian/homosexual, bisexual, questioning, queer, alternative identity, “gender questioning”: genderqueer, transgender or “other”	Inclusive policies: Legal protection against discrimination on grounds of sexual orientation and gender identity Education policies including protection for GLBTIQ students Inclusive curriculum: Useful information on homophobia/discrimination Useful information on gay, lesbian relationships, safe sex	Supportive school environments associated with supportive policies for GLBTIQ students and supportive sexuality messages Protected from discrimination Encourage wellbeing Provide messages of inclusion and affirmation Safer school environments Beneficial for all, including heterosexual students
Toomey and Russell, 2011 USA Gay-Straight Alliances, Social Justice Involvement, and School Victimization of Lesbian, Gay, Bisexual, and Queer Youth: Implications for School Well-Being and Plans to Vote.	Quantitative Paper and online survey Data from the Preventing School Harassment multi-location study (*n* = 83 schools), Total sample *n* = 1500+ Measures included: GSA presence, membership and social justice involvement	Participants (*n* = 230 youth) 12–19 years old (mean 15.7) Self-identified as lesbian, gay, bisexual, queer Participants defined their gender identity as transgender/gender queer, female, male	Gay–Straight Alliances Presence of a GSA Membership of a GSA Involvement in GSA-related social justice activities	Positively associated with school belongingness and grade point average Higher levels of personal safety at school Presence of a GSA and involvement in social justice activities buffered low levels of victimization
Toomey et al., 2011 USA High School Gay–Straight Alliances (GSAs) and Young Adult Well-Being: An Examination of GSA Presence, Participation, and Perceived Effectiveness.	Quantitative Online survey data from the wider Family Acceptance Project’s young adult survey Measures included: presence and participation in Gay–Straight Alliances (GSAs), victimization, psychological adjustment; educational outcomes	Participants (*n* = 245 youth) 21–25 years old (mean 22.1) Self-identified as lesbian, gay, bisexual and having a different sexual identity (i.e., queer, dyke or homosexual) Participants defined their gender identity as transgender, female, male	Gay–Straight Alliances Presence of a GSA GSA participation Perceived GSA effectiveness GSA presence more salient predictor of psychosocial wellbeing than membership	Significantly associated with wellbeing and self-esteem Buffered direct negative associations between LGBT victimization and depression/lifetime suicide attempts at low levels of victimization Reduced high school dropout Associated with college educational attainment
Walls et al., 2010 USA Gay-Straight Alliances and School Experiences of Sexual Minority Youth.	Quantitative Online survey data from an annual survey of the LGBT community in Colorado Measures included: presence and attendance at a GSA, school attendance, safety and presence of a safe adult	Participants (*n* = 135 youth) 13–22 years old (mean 16.3) Self-identified as lesbian, gay, bisexual, not sure, queer, other/pansexual/asexual Participants defined their gender identity as female, male, FtM trans, MtF trans	Gay–Straight Alliances: Presence of a GSA Inclusive policies: Supportive polices required to address wider school climate Provider: Supportive adult allies in the school	Increased subjective experience of student safety Increased academic achievement (grades) Increased visibility of adult allies as a resource
Lee, 2002, USA The Impact of Belonging to a High School Gay/Straight Alliance	Qualitative Face-to-face individual interviews and focus groups (alongside data from academic records andmedia and audio reports) Analysis identified the effect of GSA involvement on academic performance, relationships, being “out”, safety, contribution to society, sense of belonging	Participants (*n* = 7 youth) 15–18 years old (mean 16.0) Self-identified as: gay, lesbian, bisexual and/or straight allies All gay or lesbian participants were “out” to their parents	Gay–Straight Alliances: Presence of, and participation in, GSAs Provided opportunities to develop interpersonal relationships within school Provided opportunities for forming connections with adult mentors Provided opportunities to develop relationships with school staff	Increased school attendance, expected college attendance Perception of increased academic achievement Increased visibility, pride, openness and confidence Enhanced sense of positively contributing to society Positively impacted family and friend relationships Increased sense of school safety

**Table 7 ijerph-18-11682-t007:** Intersecting interpersonal, community and legal relations and protective factors for LGBTI+ youth wellbeing (*n* = 19 records).

Author/Year/Location Title	Methodology/Analysis	Demographic Details	Intersecting Protective Factors	Wellbeing Indicator
Paceley et al. 2020 US “It feels like home”: Transgender Youth in the Midwest and Conceptualizations of Community Climate	Qualitative In-depth interviews using community-based methods Analysis identified four themes of support for trans youth: resources, visibility, policies and ideologies	Participants (*n* = 19) 15–22 years old (mean 18.0) Self-identified as transgender man/masculine, non-binary/gender fluid, transgender woman/feminine	Interpersonal: Provider Community Groups for SGM and transgender communities and visibility Legal Presence of GSAs in schools and inclusive policies	Positive visibility A sense of belonging Personal strategies for maintaining a positive sense of self despite the potential impacts of negative climates
Wilson and Cariola, 2020 Global LGBTQI+ Youth and Mental Health: A Systematic Review of Qualitative Research.	Systematic review Search conducted across six databases Inclusion criteria: peer-reviewed journal articles in English published between 2008 and 2018, focused on LGBTQI+ youth mental health	Included studies (*n* = 34 records) LGBTQI+ participants ranged from 12 to 24 years Sample sizes ranged from *n* = 10 to *n* = 92 (with 3700 excerpts from a mixed-methods study) Qualitative (*n* = 27), mixed-methods (*n* = 7)	Interpersonal: Family and peer support and acceptance Community LGBTQI+ community-based social groups and online forums Legal Inclusive spaces in educational settings Inclusive policies/curricula/extracurricular activities	Greater self-esteem resilience Protective against depression and suicidality Significantly better psychological outcomes Comfort with sexual identity Empowerment Greater sense of school connectedness
Poštuvan et al., 2019 Global Suicidal Behaviour Among Sexual-Minority Youth: A Review of the Role of Acceptance and Support	Narrative review Search conducted across three databases Inclusion criteria: peer-reviewed journal articles in English published between 1966 and 2018, focused on LGBTI+ youth suicidality and social acceptance	Included studies LGBTI+ participants ranged from 13 to 29 years Sample sizes not available Society level, close-network level, and individual level review of acceptance and support	Interpersonal: Perceived parental support Community: Social support through connection to groups for SMY and positive media representations Legal: Protective school climates	Protective against suicidal behaviour Sense of belonging Possessing coping skills Displaying resilience Use of self-affirming strategies Involvement in activism
Taliaferro et al., 2019 USA Risk and Protective Factors for Self-Harm in a Population-Based Sample of Transgender Youth.	Quantitative In-person survey data gathered as part of a Minnesota Student Survey; total sample (*n* = 81,885) Measures included: self-harm, self-injury and suicide alongside protective factors: parent connectedness/connectedness other adults	Participants (*n* = 1635 youth) 14–17 years old (mean 15.5) Self-identified as transgender/gender nonconforming	Parent/Provider: Higher levels of connectedness to parents Connectedness to non-parental adults Community: Sense of feeling cared for by adults in community especially important Legal: Importance of school safety	Increased sense of belonging, purpose and safety Decreased odds of non-suicidal self-injury Decreased odds of suicide attempt
Eisenberg et al., 2018 US/Canada Helping Young People Stay Afloat: A Qualitative Study of Community Resources and Supports for LGBTQ Adolescents in the U.S. and Canada.	Qualitative Part of a larger mixed-methods, multisite study Go-along interviews in which participants were accompanied Analysis of organizational, community and social influences on the health and wellbeing of LGBTQ youth	Participants (*n* = 66 youth) 14–19 years old (mean 16.6) Self-identified as gay/lesbian, bisexual, trans, queer or additional or other labels including pansexual, rainbow sexual genderqueer, non-binary	Interpersonal: Family and friend support Peer/teacher allyship Community: LGBTQ youth organizations Inclusive faith communities Legal: Gay–Straight Alliances (GSAs) Inclusive policies/curricula	Connectedness with supportive peers and adults Feeling that support was consistently and easily accessible Feeling accepted and welcome Visibility and representation
Gower et al., 2018 USA Supporting Transgender and Gender Diverse Youth: Protection Against Emotional Distress and Substance Use.	Quantitative In-person survey data gathered as part of a Minnesota Student Survey; total sample (*n* = 81,885) Measures included: depression, suicidality and substance use alongside protective factors: parent connectedness/connectedness to other adults	Participants (*n* = 2168 youth) 14–17 years old (mean 15.5) Self-identified as transgender, genderqueer, genderfluid or questioning their gender	Parent: Connectedness to parents Connectedness to adult relatives Provider: Supportive teachers Community: Supportive adults in the community Legal: Feeling safe at school	Connectedness and feeling safe significantly lowered odds of emotional distress Parent connectedness protected against depression and suicidality, with a 23% reduction in the odds of depressive symptoms Feeling safe at school and connected to adults in one’s community protected against depression and suicidality
Hall, 2018 US Psychosocial Risk and Protective Factors for Depression Among Lesbian, Gay, Bisexual, andQueer Youth: A Systematic Review	Systematic review Search conducted across eight databases Inclusion criteria: quantitative published and unpublished research, since 2000, in English, on psychosocial factors and depression among LGBQ youth	Included studies (*n* = 35 records) Sample sizes ranged from *n* = 52 to *n* = 1504 LGBQ participants ranged from 15 to 24 years Journal articles (*n* = 25), theses (*n* = 9), book chapter (*n* = 1)	Interpersonal: Parents, peers (including potential romantic partners) and providers Community: LGBTI+ community connectedness—allies and supports; inclusive faith communities Legal: GSAs, inclusive policies/curricula	Perceiving LGBQ identity as positive inversely related to depression Openness about LGBQ identity inversely related to depression Affirmed LGBQ identity Promoted self-esteem
Johns et al., 2018 US Protective Factors Among Transgender and Gender Variant Youth: A Systematic Review by Socioecological Level.	Systematic review Search conducted across nineteen English and thirteen Spanish databases Inclusion criteria: peer-reviewed journal articles in English or Spanish published between 1999 and 2014 Socioecological level analysis of factors re. wellbeing	Included studies (*n* = 21 records) Sample sizes ranged from *n* = 4 to *n* = 151 (with 6803 excerpts from a mixed-methods study) Transgender and gender-variant youth participants ranged from 11 to 26 years Quantitative (*n* = 9), qualitative (*n* = 9), mixed-methods research (*n* = 3)	Interpersonal: Support of parents, peers, providers and trusted adults Community: Finding and connecting to communities of LGBTQ and allies with trans and gender-variant people Legal: GSAs, policies, curricula and availability of LGBTQ information	Enhanced self-esteem, sense of self, feelings of pride Self-advocacy, resilience and empowerment Improved mental health with fewer psychological/depressive symptoms Improved life satisfaction School safety improved attendance and aspirations
Sansfaçon et al., 2018 Canada Digging Beneath the Surface: Results from Stage One of a Qualitative Analysis of Factors Influencing the Well-Being of Trans Youth in Quebec	Qualitative Part of a Community-based Participatory Action Research project In-depth interviews Analysis identified protective factors within (1) healthcare services, (2) other institutional spaces, (3) family and (4) community spaces	Participants (*n* = 24 youth) 15–25 years (mean 20.3) Self-identified as woman or girl, trans woman, man, trans man or trans guy, straight trans man, transmasculine, masculine, non-binary, non-binary woman, non-binary trans woman, non-binary guy, masculine but fluid, transmasculine, demiboy, agender	Interpersonal: Support and acceptance of family, affirming peers, knowledgeable and accepting providers Community: LGBT community involvement Visibility of trans people/advocacy Legal: Supportive environments: health, education and work, healthcare and support related to gender identity	Acknowledgement and respect of trans youth identity Trans activism Recognition of interaction of intersectional attributes
Jones, Smith et al., 2016 Australia School Experiences of Transgender and Gender Diverse Students in Australia.	Mixed-Methods Research Online survey (*n* = 189) Online interviews (*n* = 16) Analysis identified how transgender and gender diverse students experienced school	Participants (*n* = 189 youth) 14–25 years old (mean 19.0) Self-identified as genderqueer, gender fluid, agender, trans*, androgynous, questioning, bi-gender and other, e.g., “pangender” Half (51%) defined their sexual orientation as lesbian, gay, bisexual, questioning, other or asexual	Interpersonal Use of pronouns, name and identity, access to trans-inclusive counselling, teacher support, supportive peers Community Connection with a trans community and involvement in trans activism Legal Inclusive policies Inclusive curriculum	Less likely to experience harassment/discrimination Positive feelings about gender identity, having fun and increased feelings of resilience Reduced thoughts of self-harm and suicide Prevented an act of self-harm or suicide attempt Perception of policies as protective
Snapp et al., 2015 USA Social Support Networks for LGBT Young Adults: Low Cost Strategies for Positive Adjustment	Quantitative Survey, online and paper Measures included family acceptance / support, peer support, young adult adjustment and wellbeing, life situation, self-esteem and LGBT esteem	Participants (*n* = 245 youth) 21–25 years old (mean 22.8) Self-identified a lesbian, gay, bisexual, transgender, alternative sexual identity	Parent: Family support Family acceptance had the strongest influence Peer: Friend support Community: LGBT community support	All strong predictors of positive outcomes, including life situation, self-esteem and LGBT esteem Family acceptance had the strongest overall influence Social support and sexuality-related support associated with LGBT youth wellbeing and adjustment
Higa, 2014 US Negative and Positive Factors Associated with the Well-Being of Lesbian, Gay, Bisexual, Transgender, Queer, and Questioning (LGBTQ) Youth.	Qualitative Semi-structured focus group discussion with LGBTQ youth and straight allies (*n* = 9) Individual interviews (*n* = 5) Analysis of positive factors over 8 domains	Participants (*n* = 68 youth) 14–24 years (mean 16.5) Self-identified as gay, lesbian, bisexual, transgender, queer or questioning one’s sexual or gender identity and straight allies (*n* = 11)	Interpersonal: Family, peer and mentor Community: LGBTQ communities, religious institutions, online forums, ethnic communities Legal: School climate Gay–Straight Alliances	Sense of unique LGBTQ identity Affirmation and allyship Connection, support and belonging Role model identification Visibility and acceptance
Reisner et al., 2014 USA A Compensatory Model of Risk and Resilience Applied to Adolescent Sexual Orientation Disparities in Nonsuicidal Self-Injury and Suicide Attempts.	Quantitative Survey data gathered as part of a wider Massachusetts Youth Risk Behaviour Survey; Total sample (*n* = 3131) Measures included: non-suicidal self-injury (NSSI) and suicide attempt, resilience: supportive and protective factors, risk factors	Participants (*n* = 225 youth) 14–18 years old (mean 15.5) Self-identified as lesbian, gay, bisexual and questioning LGBQ made up 7% of total sample	Parent: Family support Community: Organized afterschool, evening or weekend activities (school clubs; community centre groups, church) Legal: Extracurricular activities: music, art, dance, drama or other supervised activities	Family support was independently associated with decreased odds of both NSSSI and suicidality While school support, community engagement, sports were each not significantly protective, the number of supports was associated with decreased odds of NSSI/suicide attempt
Singh et al., 2013 US “It’s already hard enough being a student”: Developing Affirming College Environments for Trans Youth.	Qualitative Subset of a larger study examining trans youth self-advocacy and resilience Secondary analysis of semi-structured interviews Analysis identified protective factors in campus climates	Participants (*n* = 17) 15–25 years old (mean 22.0) Self-identified as guy or trans man, male, FtM, genderqueer and also queer, gay, asexual, straight	Interpersonal: Providers of trans-affirming care and staff training Community: Developing a community of trans allies on campus Legal: Inclusive policies Healthcare provision and access	Self-advocacy for the use of trans-affirming language, such as name and pronouns Enhanced resilience Sense of selves as creative agents of social change Activism for trans-affirming campus environments
Torres et al., 2012 US Examining Natural Mentoring Relationships (NMRs) Among Self-Identified Gay, Bisexual, and Questioning (GBQ) Male Youth.	Qualitative Part of a larger mixed-methods, multisite study In-depth interviews Analysis identified a diverse range of “natural mentors” and that the provision of social support was prominent in these relationships	Participants (*n* = 39 youth) 15–22 years old (mean 19.0) Self-identified as gay, bisexual and questioning (GBQ) male	Interpersonal: Providers through natural mentoring relationships Peer relationships with siblings, romantic partners and other youth Community: Visibility at events such as Pride Legal: Gay–Straight Alliances	Feeling empathic care and concern through attentive listening Enhanced coping with challenges Promoted emotional, informational, self-appraisal Feeling of unconditional support
Cohn and Hastings, 2010 US Resilience Among Rural Lesbian Youth	Narrative review No details of search strategy Inclusion criteria: literature in English, published between 1980 and 2007, focused on experiences and resilience of rural lesbian youth Analysis identified challenges that rural lesbian youth face in developing a positive self-identity including tools to enhance resilience	Included studies (19 records) Sample sizes across articles not provided Participants drawn from the Massachusetts’s Commission of Gay and Lesbian youth as aged from 14 to 25 years Journal articles (*n* = 13), book chapters (*n* = 3), reports (*n* = 2), newspaper article (*n* = 1)	Interpersonal: Consistent family support, cross-sexual orientation friendships, supportive providers: medical personnel, school staff and mental health professionals Community: Supports from organisations such as PFLAG and National Gay and Lesbian Task Force Legal: School-based supports such as Gay–Straight Alliances	Allyship and validation of non-heterosexual roles Recognition of complex social networks Visibility enhanced role model identification Potential for growth and positive development Increased resilience
Davis et al., 2009 US Supporting the Emotional and Psychological Well Being of Sexual Minority Youth: Youth Ideas for Action.	Mixed-Methods Research Secondary analysis of qualitative and quantitative data gathered through concept mapping needs assessments for two geographic communities Recruitment via two drop-in centres for GLBT and questioning youth Analysis identified 14 forms of emotional/psychological/social support for GLBT youth	Participants (*n* = 33 youth) 14–23 years old (mean 18.0) Self-identified as gay, lesbian, bisexual, transgender (GLBT) and questioning youth T and questioning youth between 14 and 23 yrs. attending drop-in centres (*n* = 33)	Interpersonal: Parental acceptance, peer support and better educated providers Community: GLBT youth space Visibility of role models, businesses and media representation Legal: Legal protection re. discrimination Positive school climate, including GSAs, inclusive school curricula, training for school personnel Access to healthcare Inclusive language, bathrooms, dress codes	Enhanced emotional wellbeing Enhanced social wellbeing Psychological and physical safety De-pathologizing GLBT identity Normalisation of GLBT identity facilitated ability to be true to self Inclusion and appreciation of within-group GLBT diversity Feeling valued and validated
Sadowski et al., 2009 US Meeting the Needs of LGBTQ Youth: A “Relational Assets” Approach.	Qualitative In-depth, open-ended interviews (*n* = 20), questionnaires (*n* = 30), cases (*n* = 3) representing a sampling of 20 youth voices Recruitment via 1 urban and 1 rural LGBTQ youth group Analysis identified relational assets in four contexts: school, family, peers and self	Participants (*n* = 30 youth) 15–22 years old (mean 19.0) All, except one, self-identified as either lesbian, gay, bisexual, transgender, queer, questioning	Interpersonal: Family relationships Peer relationships Providers: non-parent allies Community: LGBTQ youth groups and friendship networks LGBTQ role models/straight allies Legal: Gay–Straight Alliance School institutional factors	Overall feeling of connectedness to others Sense of having adult support at school Sense of the presence of allies School climate, in particular, influenced ability to make relational connections
Fenaughty and Harré, 2003 NZ Life on the Seesaw: A Qualitative Study of Suicide Resiliency Factors for Young Gay Men.	Qualitative Face-to-face interviews Analysis identified protective factors including positive social norms and conditions and high levels of support	Participants (*n* = 8 youth) 18–23 years old (mean 22.25) Self-identified as gay or queer male	Interpersonal: Family, peer and school support Availability of role models Community: LGB support groups Positive LBG representation and visibility Legal: Organised school and peer support structures Positive societal acceptance	Protective against suicidality High self-esteem Coping mechanisms Role model identification Support seeking

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
