# Peer review of "Protective Factors for LGBTI+ Youth Wellbeing: A Scoping Review Underpinned by Recognition Theory"

_ijerph, 2021, doi:10.3390/ijerph182111682_

Round 1

Reviewer 1 Report

This is a well written, comprehensive and important contribution to a field that has been heavily focussed on health disparities and difficulties encountered by LGBTI youth. It was a pleasure to read such an uplifting account of the vary many protective factors that exist within and around this population. Overall, I was very impressed with the rigour and detail provided within the review, and the synthesis of findings. I  only have some minor comments/questions that I think would improve it further. 

  • The authors have included intersex young people within the review and have provided justification for this in the introduction. However, increasingly, the needs and context for intersex youth are being noted as unique and not encompassed within the broad LGBT umbrella. See e.g. the recent Australian human rights commission report on intersex people: https://humanrights.gov.au/intersex-report-2021 It might be worth discussing this somewhat in the discussion.
  • Although the search terms are provided in the supplementary material, it would be helpful to provide a bit more detail in the search strategy section (examples perhaps) as the PCC approach is very broad and it's not clear what is/isn't included
  • The search was conducted over a year ago. While I appreciate that it would take some time to prepare this review, it could be noted in the limitations that (given the exponential increase in literature in this field) there may have been some recent relevant studies that were not included
  • I found the inclusion of section 3.4 (indicators of wellbeing) prior to description of the protective factors to be a bit confusing. To me, the indicators of wellbeing are outcomes/impacts of the protective factors so it would make sense for these to follow. If you choose to leave this section where it is, it might be helpful to provide a bit more context at the beginning of the section. 
  • In terms of stakeholder consultation, can the authors clarify whether stakeholders were presented with findings and asked for confirmation or if their thoughts were elicited without reference to findings. The way it is written, it sounds like stakeholders simply confirmed what was presented to them. 

Author Response

Attached reply submitted

Reviewer 2 Report

Thank you for sending your paper entitled “Protective Factors for LGBTI+ Youth Wellbeing: A Scoping Review Underpinned by Recognition Theory” to IJERPH. After carefully review this paper, the following comments are listed for your reference:

  1. Introduction: The introduction is well-written and clear.
  2. Materials and methods: The language of the included articles is not mentioned in the Search and study selection section, but it is specified in the Limitations section as “academic literature in English”. I would recommend to include it in the Search and study selection On the other hand, the authors stated in the Consultation sub-section that they have ethical approval; could you please provide the ethical approval number?
  3. Results: Do you mean 132 as shown in the flow diagram when you say “with 151 additional records located via...”? In the flow diagram, however, where are the “38 additional records meeting criteria”? I am not sure if the error is in the text or in the flow diagram. The explanation for the flow diagram is unclear; I would suggest a better explanation for reader understanding. The rest of the results section, on the other hand, is thoroughly explained.
  4. Discussion (921, L963): It is well-explained and precise.
  5. Conclusions (P21, L974-977): It is precise and correct.

Author Response

Attached reply submitted
